# Distributed Zero-Order Optimization under Adversarial Noise

**Arya Akhavan**
CSML, Istituto Italiano di Tecnologia
and
CREST, ENSAE, IP Paris
aria.akhavanfoomani@iit.it

**Massimiliano Pontil**
CSML, Istituto Italiano di Tecnologia
and
University College London
massimiliano.pontil@iit.it

**Alexandre B. Tsybakov**
CREST, ENSAE, IP Paris
alexandre.tsybakov@ensae.fr

## Abstract

We study the problem of distributed zero-order optimization for a class of strongly convex functions. They are formed by the average of local objectives, associated to different nodes in a prescribed network. We propose a distributed zero-order projected gradient descent algorithm to solve the problem. Exchange of information within the network is permitted only between neighbouring nodes. An important feature of our procedure is that it can query only function values, subject to a general noise model, that does not require zero mean or independent errors. We derive upper bounds for the average cumulative regret and optimization error of the algorithm which highlight the role played by a network connectivity parameter, the number of variables, the noise level, the strong convexity parameter, and smoothness properties of the local objectives. The bounds indicate some key improvements of our method over the state-of-the-art, both in the distributed and standard zero-order optimization settings. We also comment on lower bounds and observe that the dependency over certain function parameters in the bound is nearly optimal.

## 1 Introduction

We study the problem of distributed optimization where each node (or agent) has an objective function $f_i : \mathbb{R}^d \to \mathbb{R}$ and exchange of information is limited between neighbouring agents within a prescribed network of connections. The goal is to minimize the average of these objectives on a closed bounded convex set $\Theta \subset \mathbb{R}^d$,

$$\min_{x \in \Theta} f(x) \quad \text{where} \quad f(x) = \frac{1}{n} \sum_{i=1}^{n} f_i(x). \tag{1}$$

Distributed optimization has been widely studied in the literature, we refer to Tsitsiklis et al. [1986], Nedic and Ozdaglar [2009], Nedic et al. [2010], Boyd et al. [2011], Duchi et al. [2012], Jakovetić et al. [2014], Lobel et al. [2011], Kia et al. [2015], Shi et al. [2014], Jakovetić [2019], Scaman et al. [2019], Pu et al. [2021] and references therein. This problem has broad applications such as multi-agent target seeking Liu et al. [2017], distributed learning Kraska et al. [2013], and wireless networks Park et al. [2020], among others.

We address problem (1) from the perspective of zero-order distributed optimization. That is we assume that only function values can be queried by the algorithm, subject to measurement noise.

35th Conference on Neural Information Processing Systems (NeurIPS 2021).

During the optimization procedure, each agent maintains a local copy of the variables which are sequentially updated based on local and neighboring functions' queries. We wish to devise such optimization procedures which are efficient in bounding the average optimization error and cumulative regret in terms of the functions' properties and network topology.

**Contributions** Our principal contribution is a distributed zero-order optimization algorithm, introduced in Section 2, which we show to achieve tight rates of convergence under certain assumptions on the objective functions, outlined in Section 3. Specifically, we consider that the local objectives $f_i$ are $\beta$-Hölder and the average objective $f$ is $\alpha$-strongly convex. The algorithm relies on a novel zero-order gradient estimator, presented in Section 4. Although conceptually very simple, this estimator, when employed within our algorithm, allows us to obtain an $O(d^2)$ computational gain as well as improved error rates than previous state-of-the-art zero-order optimization procedures Akhavan et al. [2020], Bach and Perchet [2016], in the special case of standard (undistributed) setting. Another key advantage of our approach is due to the general noise model presented in Section 5, under which function values are queried. The noise variables do not need to be zero mean or independently sampled, and thus they include "adversarial" noise. In Section 6, we derive the rates of convergence for the cumulative regret and the optimization error of the proposed algorithm, and in Section 7 we consider the special case of 2-smooth functions. The rates highlight the dependency with respect to the number of variables $d$, the number of function queries $T$, the spectral gap of the network matrix $1 - \rho$, and the parameters $n$, $\alpha$ and $\beta$. The bounds enjoy a better dependency on $1 - \rho$ than previous bounds on zero-order distributed optimization Qu and Li [2018], Yu et al. [2019], Tang et al. [2019]. We also compare our bounds to related lower bounds in Akhavan et al. [2020] for undistributed setting, observing that our rates are optimal either with respect to $T$ and $\alpha$, or with respect to $T$ and $d$.

**Previous Work** We briefly comment on previous related work and defer to Section 8 for a more in depth discussion and comparison. For both deterministic and stochastic scenarios of problem (1), a large body of literature is devoted to first-order gradient based methods with a consensus scheme (see the papers cited above and references therein). On the other hand, the study of zero-order methods was started only recently Qu and Li [2018], Sahu et al. [2018b,a], Hajinezhad et al. [2019], Yu et al. [2019], Tang et al. [2019]. The works Qu and Li [2018], Yu et al. [2019], Tang et al. [2019] are dealing with zero-order distributed methods in noise-free settings while the noisy setting is developed in Hajinezhad et al. [2019], Sahu et al. [2018b,a]. Namely, Hajinezhad et al. [2019] considers 2-point zero-order methods with stochastic queries for non-convex optimization but assume that the noise is the same for both queries, which makes the problem analogous to noise-free scenario in terms of optimization rates. Papers Sahu et al. [2018b,a] study zero-order distributed optimization for strongly convex and $\beta$-smooth functions $f_i$ with $\beta \in \{2, 3\}$. They derive bounds on the optimization error, though without providing closed form expressions.

**Notation** Throughout we denote by $\langle \cdot, \cdot \rangle$ and $\|\cdot\|$ be the standard inner product and Euclidean norm on $\mathbb{R}^d$, respectively, and by $\|\cdot\|_*$ the spectral norm of a matrix. The notation $\mathbb{I}$ is used for the $n$-dimensional identity matrix and $\mathbb{1}$ for the vector in $\mathbb{R}^n$ with all entries equal to 1. We denote by $e_j$ the $j$-th canonical basis vector in $\mathbb{R}^d$. For any set $A$, the number of elements in $A$ is denoted by $|A|$. For $x \in \mathbb{R}$, the value $\lfloor x \rfloor$ is the maximal integer less than $x$. For every closed convex set $\Theta \subset \mathbb{R}^d$ and $x \in \mathbb{R}^d$ we denote by $\mathrm{Proj}_\Theta(x) = \mathrm{argmin}\{\|z - x\| : z \in \Theta\}$ the Euclidean projection of $x$ onto $\Theta$. We denote by $\mathrm{diam}(\Theta)$ the Euclidean diameter of $\Theta$. Finally we let $U[-1, 1]$ be the uniform distribution on $[-1, 1]$.

## 2 The Problem

Let $n$ be the number of agents and let $\mathcal{G} = (V, E)$ be an undirected graph, where $V = \{1, \ldots, n\}$ is the set of nodes and $E \subseteq V \times V$ is the set of edges. The adjacency matrix of $\mathcal{G}$ is the symmetric matrix $(A_{ij})_{i,j=1}^n$ defined as $A_{ij} = 1$, if $(i, j) \in E$ and zero otherwise. We consider the following sequential learning framework, where each agent $i$ gets values of function $f_i$ corrupted by noise and shares information with other agents. At step $t$, agent $i$ acts as follows:

- makes queries and gets noisy values of $f_i$,
- provides a local output $u^i(t)$ based on these queries and on the past information,
- broadcasts $u^i(t)$ to neighboring agents,

---

**Algorithm 1** Distributed Zero-Order Gradient

---

**Input**   Communication matrix $(W_{ij})_{i,j=1}^n$, step sizes $(\eta_t > 0)_{t=1}^{T_0-1}$
**Initialization**   Choose initial vectors $x^1(1) = \cdots = x^n(1) \in \mathbb{R}^d$
**For** $t = 1, \ldots, T_0 - 1$
   **For** $i = 1, \ldots, n$
      1.  Build an estimate $g^i(t)$ of the gradient $\nabla f_i(x^i(t))$ using noisy evaluations of $f_i$
      2.  Update $x^i(t+1) = \sum_{k=1}^n W_{ik} \mathrm{Proj}_\Theta(x^k(t) - \eta_t g^k(t))$
   **End**
**End**
**Output**   Approximate minimizer $\bar{x}(T_0) = \frac{1}{n} \sum_{i=1}^n x^i(T_0)$ of the average objective $f = \frac{1}{n} \sum_{i=1}^n f_i$

---

- updates its local variable using information from other agents as follows:

$$x^i(t+1) = \sum_{j=1}^n W_{ij} u^j(t),$$

where $W = (W_{ij})_{i,j=1}^n$ is a given matrix called the consensus matrix.

Below we use the following condition on the consensus matrix.

**Assumption A.** *Matrix $W$ is symmetric, doubly stochastic, and $\rho := \left\| W - n^{-1} \mathbb{1}\mathbb{1}^\top \right\|_* < 1$.*

Matrix $W$ accounts for the connectivity properties of the network. If $W_{ij} = 0$ the agents $i$ and $j$ are not connected (do not exchange information). Often $W$ is defined as a doubly stochastic matrix function of the adjacency matrix $A$ of the graph. One popular example is as follows:

$$W_{ij} = \begin{cases} \frac{A_{ij}}{\gamma \max\{d(i), d(j)\}} & \text{if } i \neq j, \\ 1 - \sum_{k:k\neq i} \frac{A_{ki}}{\gamma \max\{d(i), d(k)\}} & \text{if } i = j, \end{cases}$$

where $d(i) = \sum_{j=1}^n A_{ij}$ is the degree of node $i$ and $\gamma > 0$ is a constant. Then, clearly, $W = (W_{ij})$ is a symmetric and doubly stochastic matrix, and $W_{ij} = 0$ if agents $i$ and $j$ are not connected. Moreover, we have $\rho < 1 - c/n^2$ for a constant $c > 0$ (see Qu and Li [2018], Olshevsky [2014]). Values of spectral gaps $\rho$ for some other $W$ reflecting different network topologies can be found in Duchi et al. [2012]. Typically, $\rho < 1 - a_n$, where $a_n = \Omega(n^{-1})$ or $a_n = \Omega(n^{-2})$. Parameter $\rho$ can be viewed as a measure of difference between the distributed problem and a standard optimization problem. If the graph of communication is a complete graph a natural choice is $W = n^{-1} \mathbb{1}_n \mathbb{1}_n^\top$ and then $\rho = 0$. For more examples of consensus matrices $W$, see Olshevsky and Tsitsiklis [2009], Duchi et al. [2012] and references therein.

The local outputs $u^i$ can be defined in different ways. Our approach is outlined in Algorithm 1. At Step 1, an estimate of the gradient of the local objective $f_i$ at $x^i(t)$ is constructed. This involves a randomized procedure that we describe and justify in Section 4. The local output $u^i$ is defined as an update of the projected gradient algorithm with such an estimated gradient. At Step 2 of the algorithm, each agent computes the next point by a local consensus gradient descent step, which uses local and neighbor information. Step 2 of the algorithm is known as gossip method, see e.g., Boyd et al. [2006]), which was initially introduced as an approach for the networks with the imposed connection between the nodes changing by time. We also refer to Sayin et al. [2017] for similar algorithms in the context of distributed stochastic first-order gradient methods.

## 3   Assumptions on Local Objectives

In this section, we give some definitions and introduce our assumptions on the local objective functions $f_1, \ldots, f_n$.

**Definition 1.** *Denote by $\mathcal{F}_\beta(L)$ the set of all functions $f : \mathbb{R}^d \to \mathbb{R}$ that are $\ell = \lfloor \beta \rfloor$ times differentiable and satisfy, for all $x, z \in \mathbb{R}^d$ the Hölder-type condition*

$$\left| f(z) - \sum_{0 \leq |m| \leq \ell} \frac{1}{m!} D^m f(x)(z-x)^m \right| \leq L \|z - x\|^\beta, \tag{2}$$

---

**Algorithm 2** Gradient Estimator with $2d$ Queries

---

**Input**  Function $F : \mathbb{R}^d \to \mathbb{R}$ and point $x \in \mathbb{R}^d$
**Requires** Kernel $K : [-1, 1] \to \mathbb{R}$, parameter $h > 0$
**Initialization**  Generate random $r$ from uniform distribution on $[-1, 1]$
**For** $j = 1, \ldots, d$
    1.  Obtain noisy values $y_j = F(x + hre_j) + \xi_j$ and $y'_j = F(x - hre_j) + \xi'_j$
    2.  Compute $g_j = \frac{1}{2h}(y_j - y'_j)K(r)$
**End**
**Output**  $g = (g_j)_{j=1}^d \in \mathbb{R}^d$ estimator of $\nabla F(x)$

---

*where $L > 0$, the sum is over the multi-index $m = (m_1, ..., m_d) \in \mathbb{N}^d$, we used the notation $m! = m_1! \cdots m_d!$, $|m| = m_1 + \cdots + m_d$, and we defined, for every $\nu = (\nu_1, \ldots, \nu_d) \in \mathbb{R}^d$,*

$$D^m f(x)\nu^m = \frac{\partial^{|m|} f(x)}{\partial^{m_1} x_1 \cdots \partial^{m_d} x_d} \nu_1^{m_1} \cdots \nu_d^{m_d}.$$

*Elements of the class $\mathcal{F}_\beta(L)$ are referred to as $\beta$-Hölder functions.*

**Definition 2.** *Function $f : \mathbb{R}^d \to \mathbb{R}$ is called 2-smooth if it is differentiable on $\mathbb{R}^d$ and there exists $\bar{L} > 0$ such that, for every $(x, x') \in \mathbb{R}^d \times \mathbb{R}^d$, it holds that*

$$\|\nabla f(x) - \nabla f(x')\| \le \bar{L}\|x - x'\|.$$

**Definition 3.** *Let $\alpha > 0$. Function $f : \mathbb{R}^d \to \mathbb{R}$ is called $\alpha$-strongly convex if $f$ is differentiable on $\mathbb{R}^d$ and*

$$f(x) - f(x') \ge \langle \nabla f(x'), x - x' \rangle + \frac{\alpha}{2} \|x - x'\|^2, \ \forall x, x' \in \mathbb{R}^d.$$

**Assumption B.** *Functions $f_1, \ldots, f_n$: (i) belong to the class $\mathcal{F}_\beta(L)$, for some $\beta \ge 2$, and (ii) are 2-smooth.*

In Section 6 we will analyse the convergence properties of Algorithm 1 when the objective function $f$ in 1 is $\alpha$-strongly convex. We stress that we do not need the functions $f_1, \ldots, f_n$, to be as well $\alpha$-strongly convex. It is enough to make such an assumption on the compound function $f$, while the local functions $f_i$ only need to satisfy the smoothness conditions stated in Assumption B above.

## 4   Gradient Estimator

In this section, we detail our choice of gradient estimators $g^i(t)$ used at Step 1 of Algorithm 1. We consider Algorithm 2. For any function $F : \mathbb{R}^d \to \mathbb{R}$ and any point $x$, the vector $g$ returned by Algorithm 2 is an estimate of $\nabla F(x)$ based on noisy observations of $F$ at randomized points. The estimator is computed for every node $i$ at each step $t$, thus giving the vectors $g = g^i(t)$ in Algorithm 1. The gradient estimator crucially requires a kernel function $K : [-1, 1] \to \mathbb{R}$ that allows us to take advantage of possible higher order smoothness properties of $f$. Specifically, in what follows we assume that

$$\int uK(u)du = 1, \ \int u^j K(u)du = 0, \ j = 0, 2, 3, \ldots, \ell, \text{ and } \kappa_\beta \equiv \int |u|^\beta |K(u)| du < \infty, \quad (3)$$

for given $\beta \ge 2$ and $\ell = \lfloor \beta \rfloor$. In Polyak and Tsybakov [1990] such kernels can be constructed as weighted sums of Legendre polynomials, in which case $\kappa_\beta \le 2\sqrt{2}\beta$ with $\beta \ge 1$; see also Appendix A.3 in Bach and Perchet [2016] for a derivation.

The gradient estimator in Algorithm 2 differs from the standard $2d$-point Kiefer-Wolfowitz type estimator in that it uses multiplication by a random variable $K(r)$ with a well-chosen kernel $K$. On the other hand, it is also different from the previous kernel-based estimators in zero-order optimization literature Polyak and Tsybakov [1990], Bach and Perchet [2016], Akhavan et al. [2020] in that it needs $2d$ function queries per step, whereas those estimators require only one or two queries; see, in particular, Algorithm 1 in Akhavan et al. [2020] for a comparison. At first sight, this seems a big drawback of the estimator proposed here, however we will show below that thanks to this estimator

we achieve both a more efficient optimization procedure and better rate of convergences for the optimization error.

When the estimator in Algorithm 2 is used at the $t$-th outer step of Algorithm 1, it should be intended as a random variable that depends on the randomization used during the current estimation at the given node, as well as on the randomness of the past iterations, inducing the $\sigma$-algebra $\mathcal{F}_t$ (see Section 5 for the definition). Bounds for the bias of this estimator conditional on the past and for its second moment play an important role below, in our analysis of the convergence rates. These bounds are presented in the next two lemmas, whose proofs are presented in Appendix B. We state them in the simpler setting of Algorithm 2, with no reference to the filtration $(\mathcal{F}_t)_{t \in \mathbb{N}}$.

**Lemma 1.** *Let $f : \mathbb{R}^d \to \mathbb{R}$ be a function in $\mathcal{F}_\beta(L)$, $\beta \geq 2$, and let the random variables $\xi_1, \ldots, \xi_d$ and $\xi'_1, \ldots, \xi'_d$ be independent of $r$ and satisfy $\mathbb{E}[|\xi_j|] < \infty$, $\mathbb{E}[|\xi'_j|] < \infty$, for $j = 1, \ldots, d$. Let the kernel satisfy conditions* (3). *If the gradient estimator $g$ of $f$ given by Algorithm 2 then, for all $x \in \mathbb{R}^d$,*

$$\|\mathbb{E}[g] - \nabla f(x)\| \leq L\kappa_\beta \sqrt{d} h^{\beta-1}.$$

It is straightforward to see that the bound of Lemma 1 holds when the estimators are build recursively during the execution of Algorithm 1 and the expectation is taken conditionally on $\mathcal{F}_t$. This will be used in the proofs.

**Lemma 2.** *Let $f : \mathbb{R}^d \to \mathbb{R}$ be 2-smooth and let $\max_{x \in \Theta} \|\nabla f(x)\| \leq G$, $\kappa \equiv \int K^2(u) du < \infty$. Let the random variables $\xi_1, \ldots, \xi_d$ and $\xi'_1, \ldots, \xi'_d$ be independent of $r$ and $\mathbb{E}[\xi_j^2] \leq \sigma^2$, $\mathbb{E}[(\xi'_j)^2] \leq \sigma^2$ for $j = 1, \ldots, d$. If $g$ is defined by Algorithm 2, where $x$ is a random variable with values in $\Theta$ independent of $r$ and depending on $\xi_1, \ldots, \xi_d$ and $\xi'_1, \ldots, \xi'_d$ in an arbitrary way, then*

$$\mathbb{E}\|g\|^2 \leq \frac{3d\kappa}{2} \left( \frac{\sigma^2}{h^2} + \frac{3\bar{L}^2}{4} h^2 \right) + 9G^2\kappa.$$

## 5 Noise Model

Algorithm 2 is called to compute estimators of gradients of the local functions $f_i$, $i = 1, \ldots n$, at each iteration $t$ of Algorithm 1. Thus, we assume that agent $i$ at iteration $t$ generates a uniform random variable $r_i(t) \sim U[-1, 1]$ and gets $2d$ noisy observations, defined, for $j = 1, \ldots, d$

$$
\begin{aligned}
y_{i,j}(t) &= f(x^i(t) + h_t r_i(t) e_j) + \xi_{i,j}(t) \\
y'_{i,j}(t) &= f(x^i(t) + h_t r_i(t) e_j) + \xi'_{i,j}(t)
\end{aligned}
$$

where the parameters $h_t > 0$ will be specified later.

In what follows, we denote by $\mathcal{F}_t$ the $\sigma$-algebra generated by the random variables $x^i(t)$, for $i = 1, \ldots, n$. In order to meet the conditions of Lemmas 1 and 2 for each $(i, t)$, we impose the following assumption on the collection of random variables $(r_i(t), \xi_{i,j}(t), \xi'_{i,j}(t))$.

**Assumption C.** *For all integers $t$ and $i \in \{1, \ldots, n\}$ the following properties hold.*

*(i) The random variables $r_i(t) \sim U[-1, 1]$ are independent of $\xi_{i,1}(t), \ldots \xi_{i,d}(t)$, $\xi'_{i,1}(t), \ldots, \xi'_{i,d}(t)$ and from the $\sigma$-algebra $\mathcal{F}_t$,*

*(ii) $\mathbb{E}[(\xi_{i,j}(t))^2] \leq \sigma^2$, $\mathbb{E}[(\xi'_{i,j}(t))^2] \leq \sigma^2$ for $j = 1, \ldots, d$, and some $\sigma \geq 0$.*

Assumption C is very mild. Indeed, its part (i) occurs as a matter of course since it is unnatural to assume dependence between the random environment noise and artificial random variables $r_i(t)$ generated by the agents. We state (i) only for the purpose of formal rigor. Remarkably, we do not assume the noises $\xi_{i,j}(t)$ and $\xi'_{i,j}(t)$ to have zero mean. What is more, these variables can be deterministic and no independence between them for different $i, j, t$ is required, so we consider an adversarial environment. Having such a relaxed assumption on the noise is possible because of the multiplication by the zero-mean variable $K(r)$ in Algorithm 2. This and the fact that all components of the vectors are treated separately allows the proofs go through without the zero-mean assumption and under arbitrary dependence between the noises.

# 6 Main Results

In this section, we provide upper bounds on the performance of the proposed algorithms. Recall that $T_0$ is the number of outer iterations in Algorithm 1. Let $T$ be the total number of times that we observed noisy values of each $f_i$. At each iteration of Algorithm 2 we make $2d$ queries. Thus, to keep the total budget equal to $T$ we need to make $T_0 = T/(2d)$ steps of Algorithm 1 (assuming that $T/(2d)$ is an integer). We compare our results to lower bounds for any algorithm with the total budget of $T$ queries.

For given $\beta \geq 2$, we choose the tuning parameters $\eta_t$ and $h = h_t$ in Algorithms 1 and 2 as

$$\eta_t = \frac{2}{\alpha t}, \qquad \text{and} \qquad h_t = t^{-\frac{1}{2\beta}}. \tag{4}$$

Inspection of the proofs in Appendix C shows that these values of $\eta_t$ and $h_t$ lead to the best rates minimizing the bounds. As one can expect, there are two contributions to the bounds, one representing the usual stochastic optimization error, while the second one accounts for the distributed character of the problem. This second contribution to the bounds is driven by the following quantity that we call the mean discrepancy: $\Delta(t) \equiv n^{-1} \sum_{i=1}^{n} \mathbb{E}[\|x^i(t) - \bar{x}(t)\|^2]$. It plays an important role in our argument and may be of interest by itself, cf. Tang et al. [2019]. The next lemma gives a control of the mean discrepancy.

**Lemma 3.** *Let Assumptions A, B, and C hold. Let $\Theta$ be a convex compact subset of $\mathbb{R}^d$. Assume that $diam(\Theta) \leq \mathcal{K}$ and $\max_{x \in \Theta} \|\nabla f(x)\| \leq G$. If the updates $x^i(t), \bar{x}(t)$ are defined by Algorithm 1, in which the gradient estimators for $i$-th agent are defined by Algorithm 2 with $F = f_i$, $i = 1, \ldots, n$, and parameters (4) then*

$$\Delta(t) \leq \mathcal{A} \left( \frac{\rho}{1-\rho} \right)^2 \frac{d}{\alpha^2} t^{-\frac{2\beta-1}{\beta}}, \tag{5}$$

*where $\mathcal{A}$ is a constant independent of $t, d, \alpha, n, \rho$. The explicit value of $\mathcal{A}$ can be found in the proof.*

*Proof Sketch.* Let $V(t) = \sum_{i=1}^{n} \|x^i(t) - \bar{x}(t)\|^2$, and $z^i(t) = \text{Proj}_{\Theta}(x^i(t) - \eta_t g^i(t)) - (x^i(t) - \eta_t g^i(t))$. The first step is to show that, due to the definition of the algorithm and Assumptions A on matrix $W$, we have

$$V(t+1) \leq \rho^2 \sum_{i=1}^{n} \|x^i(t) - \bar{x}(t) - \eta_t(g^i(t) - \bar{g}(t)) + z^i(t) - \bar{z}(t)\|^2, \tag{6}$$

where $\bar{g}(t)$ and $\bar{z}(t)$ denote the averages of $g^i(t)$'s and $z^i(t)$'s over the agents $i$. From (6), by using the fact that $\|z^i(t)\| \leq \eta_t \|g^i(t)\|$, applying Lemma 1 conditionally on $\mathcal{F}_t$, taking expectations and then applying Lemma 2 we deduce the recursion

$$\Delta(t+1) \leq \rho \Delta(t) + \mathcal{A}_1 \frac{\rho^2}{1-\rho} \cdot \frac{d}{\alpha^2} t^{-\frac{2\beta-1}{\beta}},$$

where $\mathcal{A}_1 > 0$ is a constant. The initialization of Algorithm 1 is chosen so that $\Delta(1) = 0$. It follows that $\Delta(t)$ is bounded by a discrete convolution that can be carefully evaluated leading to (5). □

Using Lemma 3 we obtain the following theorem.

**Theorem 4.** *Let $f$ be an $\alpha$-strongly convex function and let the assumptions of Lemma 3 be satisfied. Then for any $x \in \Theta$ the cumulative regret satisfies*

$$\sum_{t=1}^{T_0} \mathbb{E}[f(\bar{x}(t)) - f(x)] \leq \frac{d}{\alpha(1-\rho)} T_0^{\frac{1}{\beta}} \left( \mathcal{B}_1 + \mathcal{B}_2 \rho^2 \right) + \frac{\mathcal{B}_3}{\alpha(1-\rho)} (\log(T_0) + 1),$$

*where the positive constants $\mathcal{B}_i$ are independent of $T_0, d, \alpha, n, \rho$. The explicit values of these constants can be found in the proof. Furthermore, if $x^*$ is the minimizer of $f$ over $\Theta$ the optimization error of the averaged estimator $\hat{x}(T_0) = \frac{1}{T_0} \sum_{t=1}^{T_0} \bar{x}(t)$ satisfies*

$$\mathbb{E}[f(\hat{x}(T_0)) - f(x^*)] \leq \frac{d}{\alpha(1-\rho)} T_0^{-\frac{\beta-1}{\beta}} \left( \mathcal{B}_1 + \mathcal{B}_2 \rho^2 \right) + \frac{\mathcal{B}_3}{\alpha(1-\rho)} \left( \frac{\log(T_0) + 1}{T_0} \right). \tag{7}$$

*Proof sketch.* Note first that, due to the definition of Algorithm 1 and to the properties of matrix $W$ we have $\bar{x}(t+1) = \bar{x}(t) - \eta_t \bar{g}(t) + \bar{z}(t)$. This resembles the usual recursion of the gradient algorithm with an additional term $\bar{z}(t) = n^{-1}\sum_{i=1}^n z^i(t)$, where $\|z^i(t)\| \le \eta_t \|g^i(t)\|$. Using this bound and $\alpha$-strong convexity of $f$, analyzing the recursion in the standard way and taking conditional expectations we obtain that, for any $x \in \Theta$,

$$f(\bar{x}(t)) - f(x) \le \frac{1}{2\eta_t}\mathbb{E}\big[a_t - a_{t+1}|\mathcal{F}_t\big] - \frac{\alpha a_t}{2} + \frac{2\eta_t}{n}\sum_{i=1}^n \mathbb{E}\big[\left\|g^i(t)\right\|^2|\mathcal{F}_t\big]$$

$$+ \underbrace{\left\|\mathbb{E}\big[\bar{g}(t)|\mathcal{F}_t\big] - \nabla f(\bar{x}(t))\right\|\|\bar{x}(t) - x\|}_{\text{Bias1}} + \underbrace{\frac{1}{\eta_t}\mathbb{E}\big[\langle\bar{z}(t), \bar{x}(t) - x\rangle|\mathcal{F}_t\big]}_{\text{Bias2}}, \qquad (8)$$

where $a_t = \|\bar{x}(t) - x\|^2$. Here, the term Bias2 is entirely induced by the distributed nature of the problem. Using the properties of Euclidean projection and some algebra, it can be bounded as

$$\text{Bias2} \le \frac{3\eta_t}{2(1-\rho)n}\sum_{i=1}^n \mathbb{E}\big[\left\|g^i(t)\right\|^2|\mathcal{F}_t\big] + \frac{1-\rho}{2n\eta_t}\sum_{i=1}^n \left\|x^i(t) - \bar{x}(t)\right\|^2. \qquad (9)$$

On the other hand, Bias1 accumulates two contributions, the first due to the gradient approximation (cf. Lemma 1) and the second due to the distributed nature of the problem:

$$\text{Bias1} \le \kappa_\beta L\sqrt{d}h_t^{\beta-1}\|\bar{x}(t) - x\| + \frac{\bar{L}}{n}\sum_{i=1}^n \left\|x^i(t) - \bar{x}(t)\right\|\|\bar{x}(t) - x\|$$

$$\le \left(\frac{(\kappa_\beta L)^2}{\alpha}dh_t^{2(\beta-1)} + \frac{\alpha a_t}{4}\right) + \left(\frac{\bar{L}t\alpha(1-\rho)}{n}\sum_{i=1}^n \left\|x^i(t) - \bar{x}\right\|^2 + \frac{\bar{L}\mathcal{K}^2}{4t\alpha(1-\rho)}\right). \quad (10)$$

Next, we combine inequalities (8)–(10), take expectations of both sides of the resulting inequality, and use Lemmas 2 and 3 to bound the second moments $\mathbb{E}\big[\left\|g^i(t)\right\|^2\big]$ and the mean discrepancy. The final result is obtained by summing up from $t = 1$ to $t = T_0$ and recalling that $\eta_t = \frac{2}{\alpha t}$, $h_t = t^{-\frac{1}{2\beta}}$. $\qquad \square$

Due to $\alpha$-strong convexity of $f$, Theorem 4 immediately implies a bound on the estimation error $\mathbb{E}[\|\hat{x}(T_0) - x^*\|^2]$. The bound is of the order of the right-hand side of (7) divided by $\alpha$. Furthermore, we get the following result about local estimators, which follows from a slight modification of Lemma 3 and Theorem 4.

**Corollary 5.** *Let Assumptions A, B, and C hold. Let $\Theta$ be a convex compact subset of $\mathbb{R}^d$. Assume that $\text{diam}(\Theta) \le \mathcal{K}$ and $\max_{x\in\Theta}\|\nabla f(x)\| \le G$. If the updates $x^i(t)$ are defined by Algorithm 1, in which the gradient estimators for $i$-th agent are defined by Algorithm 2 with $F = f_i$, $i = 1,\ldots,n$, and parameters $\eta_t = \frac{4}{\alpha(t+1)}, h_t = t^{-\frac{1}{2\beta}}$ then the local average estimator $\hat{x}^i(T_0) = \frac{2}{T_0(T_0+1)}\sum_{t=1}^{T_0} tx^i(t)$ satisfies*

$$\mathbb{E}[\|\hat{x}^i(T_0) - x^*\|^2] \le \mathcal{C}\min\left\{1, \frac{d}{\alpha^2(1-\rho)}T_0^{-\frac{\beta-1}{\beta}}\left(1 + \frac{n\rho^2}{(1-\rho)T_0}\right)\right\}, \quad i = 1,\ldots,n,$$

*where $\mathcal{C} > 0$ is a positive constant independent of $T_0, d, \alpha, n, \rho$.*

We now state a corollary of Theorem 4 for an algorithm with total budget of $T$ queries. Assume that $T_0 = T/(2d)$ is an integer. As our algorithm makes $2d$ queries per step the estimator $\hat{x}(T/(2d))$ uses the total budget of $T$ queries. Combining Theorem 4 with the trivial bound $\mathbb{E}[f(\hat{x}(T/(2d))) - f(x^*)] \le G\mathcal{K}$ we get the following result.

**Corollary 6.** *Let $T \ge 2d$ and let the assumptions of Theorem 4 be satisfied. Then we have*

$$\mathbb{E}[f(\hat{x}(T/(2d))) - f(x^*)] \le \mathcal{C}\min\left\{1, \frac{d^{2-1/\beta}}{\alpha(1-\rho)}T^{-\frac{\beta-1}{\beta}}\right\},$$

*where $\mathcal{C} > 0$ is a positive constant independent of $T, d, \alpha, n, \rho$.*

We now state several important implications of our results.

**Remark 1.** *Previous bounds on zero-order distributed optimization Qu and Li [2018], Yu et al. [2019], Tang et al. [2019] contain a dependency of $(1 - \rho)^{-2}$ in the "connectivity" parameter $\rho$. While Theorem 4 covers a more difficult noisy setting, our bound displays a better dependency of $(1 - \rho)^{-1}$. Since most common values of $1 - \rho$ are of the order $n^{-2}$ (or $n^{-1}$), this represents a substantial gain.*

**Remark 2.** *The case $n = 1$, $\rho = 0$ corresponds to usual (undistributed) zero-order stochastic optimization. Then Corollary 6 gives a bound of order $\min\left(1, \frac{d^{2-1/\beta}}{\alpha} T^{-\frac{\beta-1}{\beta}}\right)$. This improves upon the bound[1] $\min\left(1, \frac{d^2}{\alpha} T^{-\frac{\beta-1}{\beta}}\right)$ obtained under the same assumptions in Akhavan et al. [2020]. Still our bound does not match the minimax lower bound established in Akhavan et al. [2020] and equal to*

$$\min\left( \max(\alpha, T^{-1/2+1/\beta}), \frac{d}{\sqrt{T}}, \frac{d}{\alpha} T^{-\frac{\beta-1}{\beta}} \right). \tag{11}$$

*For $\alpha \asymp 1$ the lower bound (11) scales as $\min\left(1, \frac{d}{\alpha} T^{-\frac{\beta-1}{\beta}}\right)$. It has the same behavior in the interesting regime of $\alpha$ not too small ($\alpha \geq T^{-1/2+1/\beta}$) and $T \geq d$. Note, however, that the lower bound (11) is obtained for the setting with i.i.d. noise, while our upper bound is valid under adversarial noise. Therefore, it may seem rather surprising that the ratio is only $d^{1-1/\beta}$.*

**Remark 3.** *With the same budget of queries $T$, the $2d$-point method in Algorithm 2 is computationally simpler than the methods with one or two queries per step Polyak and Tsybakov [1990], Bach and Perchet [2016], Akhavan et al. [2020] previously suggested for the same setting. For example, the method in Bach and Perchet [2016], Akhavan et al. [2020] prescribes, at each step $t = 1, \ldots, T$, to generate a random variable uniformly distributed on the unit sphere in $\mathbb{R}^d$. This requires of order $d$ calls of one-dimensional random variable generator. Overall, in $T$ steps, the number of calls is of order $dT$. For our method with the same budget $T$, we make of order $T_0 = T/(2d)$ steps and at each step we need to call the generator only once in order to get $r \sim U[-1, 1]$. Thus, with the same budget of queries, Algorithm 2 needs $\sim 1/d^2$ less calls of random variable generator than the gradient estimator in Bach and Perchet [2016], Akhavan et al. [2020].*

In Appendix E, we present a numerical comparison between the algorithm proposed in this paper and that in Akhavan et al. [2020]. The results confirm our theoretical findings. The algorithm of this paper converges faster and the advantage is more pronounced as $d$ increases.

## 7 Improved Bounds for $\beta = 2$

In this section we provide improved upper bounds for the case $\beta = 2$ in Corollary 6, where we relax the dependency over $d$, from $d^{3/2}$ to $d$.

Following the literature on undistributed zero-order optimization, we use a standard 2-point method with elements of the analysis developed in Flaxman et al. [2005], Agarwal et al. [2010], Duchi et al. [2015], Shamir [2013, 2017], Akhavan et al. [2020] among others. Specifically, we define

$$g^i(t) = \frac{d}{2h_t}(y_i(t) - y_i'(t))\zeta_i(t) \tag{12}$$

$$\text{where } y_i(t) = f_i(x^i(t) + h_t\zeta_i(t)) + \xi_i(t), \quad y_i'(t) = f_i(x^i(t) - h_t\zeta_i(t)) + \xi_i'(t),$$

with the random variables $\zeta_i(t)$, $1 \leq i \leq n$, $1 \leq t \leq T$, that are i.i.d. uniformly distributed on the unit Euclidean sphere in $\mathbb{R}^d$. We make the following assumption on the noise analogous to Assumption C.

**Assumption D.** *For all integers $t$ and all $i \in \{1, \ldots, n\}$ the following properties hold.*

    *(i) The random variables $\zeta_i(t)$ are independent of $\xi_i(t)$, $\xi_i'(t)$ and from the $\sigma$-algebra $\mathcal{F}_t$,*

    *(ii) $\mathbb{E}[(\xi_i(t))^2] \leq \sigma^2$, $\mathbb{E}[(\xi_i'(t))^2] \leq \sigma^2$ for some $\sigma \geq 0$.*

**Theorem 7.** *Let $f$ be an $\alpha$-strongly convex function. Let Assumptions A, B, and D hold with $\beta = 2$. Let $\Theta$ be a convex compact subset of $\mathbb{R}^d$, and assume that $diam(\Theta) \leq \mathcal{K}$. Assume that*

---

[1]The recent work Novitskii and Gasnikov [2021] obtains the same improvement using the gradient estimator of Akhavan et al. [2020]. However, as we explain in Remark 3 that estimator is less appealing from the computational point of view.

$\max_{x \in \Theta} \|\nabla f_i(x)\| \leq G$, *for* $1 \leq i \leq n$. *Let the updates* $x^i(t), \bar{x}(t)$ *be defined by Algorithm 1, in which the gradient estimator for $i$-th agent is defined by* (12)*, and* $\eta_t = \frac{1}{\alpha t}$, $h_t = \left( \frac{3d^2 \sigma^2}{2L\alpha t + 9L^2 d^2} \right)^{1/4}$. *Then for the estimator* $\tilde{x}(T) = \frac{1}{T - \lfloor T/2 \rfloor} \sum_{t = \lfloor T/2 \rfloor + 1}^{T} \bar{x}(t)$ *we have*

$$\mathbb{E}[f(\tilde{x}(T)) - f(x^*)] \leq \frac{\mathcal{B}}{1 - \rho} \left( \frac{d}{\sqrt{\alpha T}} + \frac{d^2}{\alpha T} \right),$$

*where* $\mathcal{B} > 0$ *is a constant independent of* $T, d, \alpha, n, \rho$.

The main idea of the proof is to use surrogate functions $\hat{f}_t^i(x)$, for $1 \leq i \leq n$, defined, for every $x \in \mathbb{R}^d$, as $\hat{f}_t^i(x) = \mathbb{E} f_i(x + h_t \tilde{\zeta})$, where the expectation with respect to the random vector $\tilde{\zeta}$ uniformly distributed on the unit ball $B_d = \{u \in \mathbb{R}^d : \|u\| \leq 1\}$. A result, which can be traced back to Nemirovsky and Yudin [1983] implies the fact that $g^i(t)$ is an unbiased estimator of the gradient of the surrogate function $\hat{f}_t^i$ at $x^i(t)$. Thus, we can consider Algorithm 1 as a gradient descent for the surrogate function. Then replacing $f_i$ and $f$ by the surrogate functions with the cost of the order $h_t^2$, we can recover the initial problem. This method does not work for $\beta > 2$ since the error of approximation by surrogate function becomes of bigger order than the optimal rate $T^{-\frac{\beta - 1}{\beta}}$. The results that we implement as tools for this section are given in Appendix D.

Combining Theorem 7 with the obvious bound $\mathbb{E}[f(\tilde{x}(T)) - f(x^*)] \leq G\mathcal{K}$ we obtain

$$\mathbb{E}[f(\tilde{x}(T)) - f(x^*)] \leq \frac{\mathcal{B}'}{1 - \rho} \min \left( 1, \frac{d}{\sqrt{\alpha T}} \right), \tag{13}$$

where $\mathcal{B}' > 0$ is a constant independent of $T, d, \alpha, n, \rho$. By comparing this upper bound with the minimax lower bound (11) for $\beta = 2$, one can note that (13) is optimal with respect to the parameters $T$ and $d$ when $\alpha \asymp 1$.

## 8   Discussion

We expand our discussion on previous related work, comparing our results to the state-of-the-art distributed and undistributed zero-order optimization settings, and highlight few key open problems.

**Comparison to Zero-Order Distributed Settings**   Distributed opimization with noisy functions' queries was considered in detail in Sahu et al. [2018b,a], where the setting differs from ours in some key aspects: the updates are obtained not as in Step 2 of Algorithm 1 but rather via decentralized techniques, matrix $W$ is random, the noise is zero-mean random rather than adversarial, and 2-point gradient estimator is used. Papers Sahu et al. [2018b,a] provide, for $\beta = 2$ and $\beta = 3$, bounds on $\mathbb{E}[\|x^i(T) - x^*\|^2]$ of the order at least $\frac{n^{3/2}}{(1 - \rho)^2} T^{-1/2}$ and $\frac{n^{3/2}}{(1 - \rho)^2} T^{-2/3}$, respectively, as functions of $n, \rho$ and $T$. Their bounds contain uncontrolled terms of the form $\mathbb{E}[\|x^i(k_0) - x^*\|^2]$ for some large enough $k_0 = k_0(n, \alpha, d)$ leaving unclear the resulting rate. Paper Hajinezhad et al. [2019] considers 2-point methods with stochastic queries but assume that the noise is the same for both queries and deal with non-convex optimization. Noisy-free zero-order distributed optimization is studied by Qu and Li [2018], Yu et al. [2019], Tang et al. [2019]. From these, Tang et al. [2019] is the closest to our work as it builds on the updates as at Step 2 of Algorithm 1 (though without projections). The bounds obtained therein are of the order $(1 - \rho)^{-2}$ considered as functions of $\rho$, although they hold for the larger class of gradient dominant functions. As noted in Remark 1 the bound of Theorem 4 scales only as $(1 - \rho)^{-1}$ and this bound holds true, in particular, for noisy-free setting, which is its special case corresponding to $\sigma = 0$. Since most common values of $1 - \rho$ are of the order $n^{-2}$ (or $n^{-1}$), this represents a substantial gain. Moreover, Theorem 4 covers a difficult noise setting as we deal with adversarial noise. It is also worthwhile to note that the first-order distributed optimization exhibits much better dependency on $\rho$ since bounds that scale as $(1 - \rho)^{-1/2}$ can be achieved, see [Duchi et al., 2015, Scaman et al., 2019]. Some of the references mentioned above considered unconstrained optimization while our results deal with constrained optimization. Note that the only difference in the proofs of the upper bounds for constrained and unconstrained cases is in the presence of an additional term proportional to the second moment of the gradient at the update (see Lemma 2.4 in Akhavan et al. [2020] for a similar argument). Since this additional term is bounded independently of $\rho$ the overall dependency on $\rho$ remains the same.

**Computational and Statistical Advantage of the Proposed Gradient Estimator** As we high-lighted in Section 6 the gradient estimator in Algorithm 2 requires $2d$ function queries. At first sight this seems problematic when the dimension $d$ is high, as they need at least $T = 2d$ queries. However, the lower bounds in Shamir [2013], Akhavan et al. [2020] reported in (11) above indicate that no estimator can achieve nontrivial convergence rate for zero-order optimization when $T \lesssim d^{\frac{\beta}{\beta-1}}$. Thus, having the total budget of $T \gg d$ queries is a necessary condition for success of any zero-order stochastic optimization method. Algorithms with one or two queries per step can, of course, be realized for $T \lesssim d$ but in this case they do not enjoy any nontrivial error behavior. Moreover, by Remark 3, with the same total budget of queries $T$, the gradient estimator from Algorithm 2 is computationally more efficient[2] than the estimators in Polyak and Tsybakov [1990], Bach and Perchet [2016], Akhavan et al. [2020]. Indeed, with the same budget of queries, it needs $1/d^2$ less calls of random variable generator than it would be for the gradient estimator in Bach and Perchet [2016], Akhavan et al. [2020]. At the same time, as detailed in Remark 2 the proposed gradient estimator yields better rates on the optimization error. We conclude that the proposed zero-order optimization procedure provides both a computational and statistical improvement over the state-of-the-art methods in Akhavan et al. [2020].

**Limitations and Future Work** A main problem, which remains open, is to study whether the dependency of $(1-\rho)^{-1}$ in the upper bounds in Corollary 6 and Theorem 7 is minimax optimal. Moreover, in the standard (undistributed) setting it remains an open problem to design a zero-order optimization procedure that meets the minimax lower bound (11) with respect to all problem parameters ($T, d, \beta$ and $\alpha$). Further directions of research include the analysis of disturbed zero-order algorithms for larger classes of functions, such as $\alpha$-gradient dominant ones, as well as extension of our results to stochastic updates or asynchronous activation schemes.

# Funding Disclosure

A. Akhavan and M. Pontil were partially supported by SAP SE. The research of A.B. Tsybakov is supported by a grant of the French National Research Agency (ANR), "Investissements d'Avenir" (LabEx Ecodec/ANR-11-LABX-0047).

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
