## A  Auxiliary Lemma

**Lemma 8.** *Let $W$ be a matrix satisfying Assumption A and let $x^i = \sum_{j=1}^n W_{i,j} u^j$ for $i = 1, \ldots, n$, where $u^1, \ldots, u^n$ are some vectors in $\mathbb{R}^d$. Set $\bar{x} = n^{-1} \sum_{i=1}^n x^i$, $\bar{u} = n^{-1} \sum_{i=1}^n u^i$. Then*

$$\sum_{i=1}^n \left\| x^i - \bar{x} \right\|^2 \le \rho^2 \sum_{i=1}^n \left\| u^i - \bar{u} \right\|^2.$$

*Proof.* Introduce the matrices $X^\top = (x^1, \ldots, x^n) \in \mathbb{R}^{d \times n}$, $U^\top = (u^1, \ldots, u^n) \in \mathbb{R}^{d \times n}$ and the centering matrix $H = \mathbb{I} - \frac{1}{n} \mathbb{1}\mathbb{1}^\top \in \mathbb{R}^{n \times n}$. Notice that $\sum_{i=1}^n \left\| x^i - \bar{x} \right\|^2 = \mathrm{Tr}(\Sigma)$, where $\mathrm{Tr}(\Sigma)$ is the trace of the matrix

$$\Sigma = \sum_{i=1}^n (x^i - \bar{x})(x^i - \bar{x})^\top = \sum_{i=1}^n x^i (x^i)^\top - \bar{x}\bar{x}^\top = X^\top H X.$$

It is not hard to check that $\mathrm{Tr}(\Sigma) = \mathrm{Tr}(U^\top W H W U)$. Moreover, as $W$ is symmetric and $W\mathbb{1} = \mathbb{1}$ we have $HW = W - \frac{1}{n}\mathbb{1}\mathbb{1}^\top := \overline{W} = WH$. Thus, $WHW = WH^2 W = H\overline{W}^2 H$ and

$$\mathrm{Tr}(\Sigma) = \mathrm{Tr}(U^\top H \overline{W}^2 H U) \le \|\overline{W}^2\|_* \, \mathrm{Tr}(U^\top H^2 U) \le \rho^2 \mathrm{Tr}(U^\top H U) = \rho^2 \sum_{i=1}^n \left\| u^i - \bar{u} \right\|^2.$$

$\square$

## B  Proofs for Section 4

**Lemma 1.** *Let $f : \mathbb{R}^d \to \mathbb{R}$ be a function in $\mathcal{F}_\beta(L)$, $\beta \ge 2$, and let the random variables $\xi_1, \ldots, \xi_d$ and $\xi_1', \ldots, \xi_d'$ be independent of $r$ and satisfy $\mathbb{E}[|\xi_j|] < \infty$, $\mathbb{E}[|\xi_j'|] < \infty$, for $j = 1, \ldots, d$. Let the kernel satisfy conditions (3). If the gradient estimator $g$ of $f$ given by Algorithm 2 then, for all $x \in \mathbb{R}^d$,*

$$\|\mathbb{E}[g] - \nabla f(x)\| \le L \kappa_\beta \sqrt{d} h^{\beta - 1}.$$

*Proof.* By Taylor expansion we have

$$\frac{f(x + h r e_j) - f(x - h r e_j)}{2h} = \frac{\partial f(x)}{\partial x_j} r + \frac{1}{h} \sum_{2 \le m \le \ell,\, m \text{ odd}} \frac{(rh)^m}{m!} \frac{\partial^m f(x)}{\partial x_j^m} + \frac{R(h r e_j) - R(-h r e_j)}{2h},$$

where $|R(\pm h r e_j)| \le L \|h r e_j\|^\beta = L|r|^\beta h^\beta$. Using (3) it follows that

$$\left| \mathbb{E}[g_j] - \frac{\partial f(x)}{\partial x_j} \right| = \left| \mathbb{E}\left[ \frac{f(x + h r e_j) - f(x - h r e_j)}{2h} K(r) \right] - \frac{\partial f(x)}{\partial x_j} \right| \le L \kappa_\beta h^{\beta - 1},$$

which implies the result. $\square$

**Lemma 2.** *Let $f : \mathbb{R}^d \to \mathbb{R}$ be 2-smooth and let $\max_{x \in \Theta} \|\nabla f(x)\| \le G$, $\kappa \equiv \int K^2(u) du < \infty$. Let the random variables $\xi_1, \ldots, \xi_d$ and $\xi_1', \ldots, \xi_d'$ be independent of $r$ and $\mathbb{E}[\xi_j^2] \le \sigma^2$, $\mathbb{E}[(\xi_j')^2] \le \sigma^2$ for $j = 1, \ldots, d$. If $g$ is defined by Algorithm 2, where $x$ is a random variable with values in $\Theta$ independent of $r$ and depending on $\xi_1, \ldots, \xi_d$ and $\xi_1', \ldots, \xi_d'$ in an arbitrary way, then*

$$\mathbb{E}\|g\|^2 \le \frac{3 d \kappa}{2} \left( \frac{\sigma^2}{h^2} + \frac{3 \bar{L}^2}{4} h^2 \right) + 9 G^2 \kappa.$$

*Proof.* Fix $j \in 1, \ldots, d$. Using the inequality $(a + b + c)^2 \le 3(a^2 + b^2 + c^2)$ and the independence between $r$ and $(\xi_j, \xi_j')$ we have

$$\begin{aligned}
\mathbb{E}[g_j^2] &= \frac{1}{4h^2} \mathbb{E}\left[ (f(x + h r e_j) - f(x - h r e_i) + \xi_i - \xi_i')^2 K^2(r) \right] \qquad (14) \\
&\le \frac{3}{4h^2} \mathbb{E}\left[ \left( (f(x + h r e_j) - f(x - h r e_j))^2 + 2\sigma^2 \right) K^2(r) \right].
\end{aligned}$$

The same calculations as in the proof of Lemma 2.4 in Akhavan et al. [2020] yield

$$\left(f(x + hre_j) - f(x - hre_j)\right)^2 \leq 3\left(\frac{\bar{L}^2}{2}\|hre_j\|^4 + 4\langle \nabla f(x), hre_j \rangle^2\right),$$

Finally, we combine this inequality with (14) to obtain

$$\mathbb{E}[g_j^2] \leq \frac{3}{2}\kappa\left(\frac{\sigma^2}{h^2} + \frac{3\bar{L}^2}{4}h^2\right) + 9\kappa\mathbb{E}[\langle \nabla f(x), e_i \rangle^2],$$

which immediately implies the lemma. $\qquad\square$

## C    Proofs for Section 6

Recall the notation $\Delta(t) = n^{-1}\sum_{i=1}^n \mathbb{E}[\|x^i(t) - \bar{x}(t)\|^2]$, $\bar{g}(t) = \frac{1}{n}\sum_{i=1}^n g^i(t)$, and $z^i(t) = \text{Proj}_\Theta\left(x^i(t) - \eta_t g^i(t)\right) - (x^i(t) - \eta_t g^i(t))$. We also set $\bar{z}(t) = \frac{1}{n}\sum_{i=1}^n z^i(t)$.

**Lemma 3.** *Let Assumptions A, B, and C hold. Let $\Theta$ be a convex compact subset of $\mathbb{R}^d$. Assume that $\text{diam}(\Theta) \leq \mathcal{K}$ and $\max_{x\in\Theta}\|\nabla f(x)\| \leq G$. If the updates $x^i(t), \bar{x}(t)$ are defined by Algorithm 1, in which the gradient estimators for $i$-th agent are defined by Algorithm 2 with $F = f_i$, $i = 1, \ldots, n$, and parameters (4) then*

$$\Delta(t) \leq \mathcal{A}\left(\frac{\rho}{1-\rho}\right)^2 \frac{d}{\alpha^2} t^{-\frac{2\beta-1}{\beta}}, \tag{5}$$

*where $\mathcal{A}$ is a constant independent of $t, d, \alpha, n, \rho$. The explicit value of $\mathcal{A}$ can be found in the proof.*

*Proof.* Set $V(t) = \sum_{i=1}^n \|x^i(t) - \bar{x}(t)\|^2$. The definition of Algorithm 1 and Lemma 8 imply:

$$V(t+1) \leq \rho^2 \sum_{i=1}^n \|x^i(t) - \bar{x}(t) - \eta_t(g^i(t) - \bar{g}(t)) + z^i(t) - \bar{z}(t)\|^2.$$

The result is immediate if $\rho = 0$. Therefore, in rest of the proof we assume that $\rho > 0$. We have

$$V(t+1) \leq \rho^2 \sum_{i=1}^n \Big[V(t) + \eta_t^2 \|g^i(t) - \bar{g}(t)\|^2 + \|z^i(t) - \bar{z}(t)\|^2 \tag{15}$$

$$- 2\eta_t\Big\langle x^i(t) - \bar{x}(t), g^i(t) - \bar{g}(t)\Big\rangle \tag{16}$$

$$- 2\eta_t\Big\langle g^i(t) - \bar{g}(t), z^i(t) - \bar{z}(t)\Big\rangle \tag{17}$$

$$+ 2\Big\langle x^i(t) - \bar{x}(t), z^i(t) - \bar{z}(t)\Big\rangle\Big]. \tag{18}$$

For any $z \in \mathbb{R}^d$, we have $\sum_{i=1}^n \|g^i(t) - \bar{g}(t)\|^2 \leq \sum_{i=1}^n \|g^i(t) - z\|^2$, so that

$$\eta_t^2 \sum_{i=1}^n \mathbb{E}\big[\|g^i(t) - \bar{g}(t)\|^2 \,|\mathcal{F}_t\big] \leq \eta_t^2 \sum_{i=1}^n \mathbb{E}\big[\|g^i(t)\|^2 \,|\mathcal{F}_t\big].$$

Next, from the definition of the projection,

$$\|z^i(t)\| = \left\|\text{Proj}_\Theta\left(x^i - \eta_t g^i(t)\right) - (x^i - \eta_t g^i(t))\right\|$$

$$\leq \|x^i - (x^i - \eta_t g^i(t))\| = \eta_t \|g^i(t)\|. \tag{19}$$

Therefore, for the term containing $\|z^i(t) - \bar{z}(t)\|^2$ in (15) we obtain

$$\sum_{i=1}^n \mathbb{E}[\|z^i(t) - \bar{z}(t)\|^2 \,|\mathcal{F}_t] \leq \sum_{i=1}^n \mathbb{E}[\|z^i(t)\|^2 \,|\mathcal{F}_t] \leq \eta_t^2 \sum_{i=1}^n \mathbb{E}\Big[\|g^i(t)\|^2 \,|\mathcal{F}_t\Big].$$

For the expression in (16), by decoupling we get

$$-2\eta_t \sum_{i=1}^n \mathbb{E}\Big[\Big\langle x^i(t) - \bar{x}(t), g^i(t) - \bar{g}(t)\Big\rangle|\mathcal{F}_t\Big] \leq \lambda V(t) + \frac{\eta_t^2}{\lambda} \sum_{i=1}^n \mathbb{E}\Big[\big\|g^i(t)\big\|^2 |\mathcal{F}_t\Big],$$

where $\lambda > 0$ is a value to be chosen later. For the expression in (17), we have

$$-2\eta_t \sum_{i=1}^n \mathbb{E}\Big[\Big\langle g^i(t) - \bar{g}(t), z^i(t) - \bar{z}(t)\Big\rangle|\mathcal{F}_t\Big] \leq \eta_t^2 \sum_{i=1}^n \mathbb{E}\Big[\big\|g^i(t) - \bar{g}(t)\big\|^2 |\mathcal{F}_t\Big] + \sum_{i=1}^n \mathbb{E}\Big[\big\|z^i(t) - \bar{z}(t)\big\|^2 |\mathcal{F}_t\Big]$$

$$\leq 2\eta_t^2 \sum_{i=1}^n \mathbb{E}\Big[\big\|g^i(t)\big\|^2 |\mathcal{F}_t\Big].$$

Similarly, for the expression in (18), using the Cauchy–Schwarz inequality we get

$$2\sum_{i=1}^n \mathbb{E}\Big[\Big\langle x^i(t) - \bar{x}(t), z^i(t) - \bar{z}(t)\Big\rangle|\mathcal{F}_t\Big] \leq 2\sum_{i=1}^n \mathbb{E}\Big[\big\|x^i(t) - \bar{x}(t)\big\|\,\big\|z^i(t) - \bar{z}(t)\big\|\,|\mathcal{F}_t\Big]$$

$$\leq \lambda V(t) + \frac{1}{\lambda}\sum_{i=1}^n \mathbb{E}\Big[\big\|z^i(t) - \bar{z}(t)\big\|^2 |\mathcal{F}_t\Big]$$

$$\leq \lambda V(t) + \frac{\eta_t^2}{\lambda}\sum_{i=1}^n \mathbb{E}\Big[\big\|g^i(t)\big\|^2 |\mathcal{F}_t\Big].$$

Combining the above inequalities yields

$$\mathbb{E}[V(t+1)|\mathcal{F}_t] \leq \rho^2(1+2\lambda)V(t) + \rho^2\Big(4 + \frac{2}{\lambda}\Big)\eta_t^2 \sum_{i=1}^n \mathbb{E}\Big[\big\|g^i(t)\big\|^2 |\mathcal{F}_t\Big]. \tag{20}$$

Taking expectations in (20) and applying Lemma 2 we obtain

$$\Delta(t+1) \leq \rho^2(1+2\lambda)\Delta(t) + \rho^2\Big(4 + \frac{2}{\lambda}\Big)\eta_t^2\Big(9\kappa G^2 + d\Big(\frac{9h_t^2\kappa\bar{L}^2}{8} + \frac{3\kappa\sigma^2}{2h_t^2}\Big)\Big).$$

Choose here $\lambda = \frac{1-\rho}{2\rho}$. Then, using the fact that $\eta_t = \frac{2}{\alpha t}$, $h_t = t^{-\frac{1}{2\beta}}$ we find

$$\Delta(t+1) \leq \rho\Delta(t) + \mathcal{A}_1 \frac{\rho^2}{1-\rho} \cdot \frac{d}{\alpha^2} t^{-\frac{2\beta-1}{\beta}}, \tag{21}$$

where $\mathcal{A}_1 = \frac{144\kappa G^2}{d} + 18\kappa\bar{L}^2 + 24\kappa\sigma^2$. Due to the recursion in (21) we have, for any $t \geq 3$,

$$\Delta(t+1) \leq \rho^t\Delta(1) + \mathcal{A}_1 \frac{\rho^2}{1-\rho} \cdot \frac{d}{\alpha^2} \sum_{s=1}^t s^{-\frac{2\beta-1}{\beta}}\rho^{t-s}$$

$$\leq \mathcal{A}_1 \frac{\rho^2}{1-\rho} \cdot \frac{d}{\alpha^2}\Big(\frac{1}{\lfloor\frac{t}{2}\rfloor}\sum_{s=1}^{\lfloor\frac{t}{2}\rfloor} s^{-\frac{2\beta-1}{\beta}} \sum_{k=t-\lfloor\frac{t}{2}\rfloor}^{t-1}\rho^k + \frac{1}{\lfloor\frac{t}{2}\rfloor}\sum_{s=\lfloor\frac{t}{2}\rfloor+1}^t s^{-\frac{2\beta-1}{\beta}} \sum_{k=0}^{t-\lfloor\frac{t}{2}\rfloor-1}\rho^k\Big), \tag{22}$$

where $\Delta(1) = 0$ by the choice of initial values and the last inequality uses the fact that if the function $\phi_1(\cdot)$ is monotone decreasing and $\phi_2(\cdot)$ is monotone increasing then

$$\frac{1}{S}\sum_{s=1}^S \phi_1(s)\phi_2(s) \leq \Big(\frac{1}{S}\sum_{s=1}^S \phi_1(s)\Big)\Big(\frac{1}{S}\sum_{s=1}^S \phi_2(s)\Big),$$

see, e.g., [Devroye et al., 1996, Theorem A.19]. The sums in (22) satisfy

$$\sum_{s=1}^{\lfloor\frac{t}{2}\rfloor} s^{-\frac{2\beta-1}{\beta}} \leq 1 + \int_1^\infty s^{-\frac{2\beta-1}{\beta}} = \frac{2\beta-1}{\beta-1}, \qquad \sum_{s=\lfloor\frac{t}{2}\rfloor+1}^t s^{-\frac{2\beta-1}{\beta}} \leq \frac{t}{2}\Big(\frac{t}{2}\Big)^{-\frac{2\beta-1}{\beta}} = 2^{\frac{\beta-1}{\beta}}t^{-\frac{\beta-1}{\beta}},$$

$$\sum_{k=0}^{t-\lfloor\frac{t}{2}\rfloor-1} \rho^k \leq \frac{1}{1-\rho}, \qquad \sum_{k=t-\lfloor\frac{t}{2}\rfloor}^{t-1} \rho^k \leq \sum_{k=\lfloor\frac{t}{2}\rfloor}^{t-1} \rho^k \leq t\rho^{\lfloor\frac{t}{2}\rfloor}/2 \leq \frac{8}{\log(1/\rho)t},$$

where the last inequality follows from the fact that $\rho^k \leq \frac{1}{\log(1/\rho)k^2}$ for any positive integer $k$. Plugging the above inequalities in (22) gives

$$\Delta(t+1) \leq \mathcal{A}_1 \frac{\rho^2}{1-\rho}\frac{d}{\alpha^2}\left(\frac{24}{\log(1/\rho)t^2}\frac{2\beta-1}{\beta-1} + 3(2^{\frac{\beta-1}{\beta}})\frac{t^{-\frac{2\beta-1}{\beta}}}{1-\rho}\right)$$

$$\leq \mathcal{A}_2 \frac{\rho^2}{(1-\rho)^2}\frac{d}{\alpha^2}t^{-\frac{2\beta-1}{\beta}},$$

where $\mathcal{A}_2 = \left(24\frac{2\beta-1}{\beta-1} + 3(2^{\frac{\beta-1}{\beta}})\right)\mathcal{A}_1$. Therefore, setting $\mathcal{A} := 2\mathcal{A}_2$ we conclude that, for $t \geq 3$,

$$\Delta(t) \leq \mathcal{A}\frac{\rho^2}{(1-\rho)^2}\frac{d}{\alpha^2}t^{-\frac{2\beta-1}{\beta}}.$$

For $t \in \{1,2\}$ the bound of the lemma holds trivially since $\bar{x}$ and all $x^i$ belong to the compact $\Theta$.

$\square$

**Theorem 4.** *Let $f$ be an $\alpha$-strongly convex function and let the assumptions of Lemma 3 be satisfied. Then for any $x \in \Theta$ the cumulative regret satisfies*

$$\sum_{t=1}^{T_0} \mathbb{E}\big[f(\bar{x}(t)) - f(x)\big] \leq \frac{d}{\alpha(1-\rho)}T_0^{\frac{1}{\beta}}\left(\mathcal{B}_1 + \mathcal{B}_2\rho^2\right) + \frac{\mathcal{B}_3}{\alpha(1-\rho)}(\log(T_0) + 1),$$

*where the positive constants $\mathcal{B}_i$ are independent of $T_0, d, \alpha, n, \rho$. The explicit values of these constants can be found in the proof. Furthermore, if $x^*$ is the minimizer of $f$ over $\Theta$ the optimization error of the averaged estimator $\hat{x}(T_0) = \frac{1}{T_0}\sum_{t=1}^{T_0}\bar{x}(t)$ satisfies*

$$\mathbb{E}[f(\hat{x}(T_0)) - f(x^*)] \leq \frac{d}{\alpha(1-\rho)}T_0^{-\frac{\beta-1}{\beta}}\left(\mathcal{B}_1 + \mathcal{B}_2\rho^2\right) + \frac{\mathcal{B}_3}{\alpha(1-\rho)}\left(\frac{\log(T_0)+1}{T_0}\right). \quad (7)$$

*Proof.* From the definition of Algorithm 1 and (19) we obtain

$$\|\bar{x}(t+1) - x\|^2 = \|\bar{x}(t) - x\|^2 + \|\bar{z}(t)\|^2 + \eta_t^2\|\bar{g}(t)\|^2$$
$$- 2\eta_t\langle\bar{g}(t), \bar{x}(t) - x\rangle + 2\langle\bar{z}(t), \bar{x}(t) - x\rangle - 2\eta_t\langle\bar{z}(t), \bar{g}(t)\rangle$$

$$\leq \|\bar{x}(t) - x\|^2 - 2\eta_t\langle\bar{g}(t), \bar{x}(t) - x\rangle + 2\langle\bar{z}(t), \bar{x}(t) - x\rangle + \frac{4\eta_t^2}{n}\sum_{i=1}^n\|g^i(t)\|^2.$$

It follows that

$$\langle\bar{g}(t), \bar{x}(t) - x\rangle \leq \frac{\|\bar{x}(t) - x\|^2 - \|\bar{x}(t+1) - x\|^2}{2\eta_t} + \frac{1}{\eta_t}\langle\bar{z}(t), \bar{x}(t) - x\rangle + \frac{2\eta_t}{n}\sum_{i=1}^n\|g^i(t)\|^2.$$

The strong convexity assumption implies

$$f(\bar{x}(t)) - f(x) \leq \langle\nabla f(\bar{x}(t)), \bar{x}(t) - x\rangle - \frac{\alpha}{2}\|\bar{x}(t) - x\|^2.$$

Combining the last two displays and taking conditional expectations from both sides we get

$$\mathbb{E}\big[f(\bar{x}(t)) - f(x)|\mathcal{F}_t\big] \leq \big\|\mathbb{E}\big[\bar{g}(t)|\mathcal{F}_t\big] - \nabla f(\bar{x}(t))\big\|\,\|\bar{x}(t) - x\| + \frac{1}{2\eta_t}\mathbb{E}\big[a_t - a_{t+1}|\mathcal{F}_t\big]$$

$$+ \frac{2\eta_t}{n}\sum_{i=1}^n\mathbb{E}\big[\|g^i(t)\|^2|\mathcal{F}_t\big] - \frac{\alpha}{2}a_t + \frac{1}{\eta_t}\mathbb{E}\big[\langle\bar{z}(t), \bar{x}(t) - x\rangle|\mathcal{F}_t\big], \quad (23)$$

where $a_t = \|\bar{x}(t) - x\|^2$.

The first term in right hand side of (23) is bounded as follows

$$
\begin{aligned}
\left\|\mathbb{E}\big[\bar{g}(t)|\mathcal{F}_t\big] - \nabla f(\bar{x}(t))\right\| \|\bar{x}(t) - x\| &\leq \left[\left\|\mathbb{E}\big[\bar{g}(t)|\mathcal{F}_t\big] - \frac{1}{n}\sum_{i=1}^{n}\nabla f_i(x^i(t))\right\|\right.\\
&+ \left.\left\|\frac{1}{n}\sum_{i=1}^{n}\nabla f_i(x^i(t)) - \frac{1}{n}\sum_{i=1}^{n}\nabla f_i(\bar{x}(t))\right\|\right] \|\bar{x}(t) - x\|\\
&\leq \kappa_\beta L\sqrt{d}h_t^{\beta-1}\|\bar{x}(t) - x\| + \frac{\bar{L}}{n}\sum_{i=1}^{n}\left\|x^i(t) - \bar{x}(t)\right\| \|\bar{x}(t) - x\|,
\end{aligned}
\tag{24}
$$

where the last inequality is due to Lemma 1 and Assumption B(ii). We now decouple the terms in (24) using the fact that $ab \leq \frac{a^2}{v} + \frac{vb^2}{4}, \forall a, b \geq 0, v > 0$. Thus, we obtain

$$
\kappa_\beta L\sqrt{d}h_t^{\beta-1}\|\bar{x}(t) - x\| \leq \frac{(\kappa_\beta L)^2}{\alpha}dh_t^{2(\beta-1)} + \frac{\alpha}{4}\|\bar{x}(t) - x\|^2
\tag{25}
$$

and, taking $v = t\alpha(1 - \rho)$,

$$
\frac{\bar{L}}{n}\sum_{i=1}^{n}\left\|x^i(t) - \bar{x}(t)\right\| \|\bar{x}(t) - x\| \leq \frac{\bar{L}t\alpha(1-\rho)}{n}\sum_{i=1}^{n}\left\|x^i(t) - \bar{x}\right\|^2 + \frac{\bar{L}\mathcal{K}^2}{4t\alpha(1-\rho)}.
\tag{26}
$$

The bound (26) brings us to the quantity $\sum_{i=1}^{n}\left\|x^i(t) - \bar{x}(t)\right\|^2$ that can be controlled in expectation via Lemma 3. Note that the choice of $v = t\alpha(1 - \rho)$ here is motivated by the fact that, once Lemma 3 is applied (see the end of this proof), it minimizes the final bound in $\rho$ and $\alpha$. We could have kept $v$ in the form $v = v_0 t$ (with an arbitrary parameter $v_0 > 0$) until the application of Lemma 3 and then optimize over $v_0$. However, we prefer to insert the optimal value $v_0 = \alpha(1 - \rho)$ already at this stage.

Combining (25) and (26) with (24) gives

$$
\begin{aligned}
\left\|\mathbb{E}\big[\bar{g}(t)|\mathcal{F}_t\big] - \nabla f(\bar{x}(t))\right\| \|\bar{x}(t) - x\| \leq& \frac{(\kappa_\beta L)^2}{\alpha}dh_t^{2(\beta-1)} + \frac{\alpha}{4}\|\bar{x}(t) - x\|^2 +\\
&+ \frac{\bar{L}t\alpha(1-\rho)}{n}\sum_{i=1}^{n}\left\|x^i(t) - \bar{x}(t)\right\|^2 + \frac{\bar{L}\mathcal{K}^2}{4t\alpha(1-\rho)}.
\end{aligned}
\tag{27}
$$

Next, we have

$$
\begin{aligned}
\frac{1}{\eta_t}\langle\bar{z}(t), \bar{x}(t) - x\rangle &= \frac{1}{n\eta_t}\sum_{i=1}^{n}\langle z^i(t), \bar{x}(t) - x\rangle\\
&\leq \frac{1}{n\eta_t}\sum_{i=1}^{n}\langle z^i(t), \bar{x}(t) - \big(x^i(t) - \eta_t g^i(t)\big)\rangle + \langle z^i(t), \big(x^i(t) - \eta_t g^i(t)\big) - x\rangle.
\end{aligned}
\tag{28}
$$

Since $\text{Proj}_\Theta(\cdot)$ is the Euclidean projection on the convex set $\Theta$, for any $w \in \mathbb{R}^d, x \in \Theta$ we have $\langle\text{Proj}_\Theta(w) - w, \text{Proj}_\Theta(w) - x\rangle \leq 0$, which implies

$$
\langle\text{Proj}_\Theta(w) - w, w - x\rangle = -\left\|\text{Proj}_\Theta(w) - w\right\|^2 + \langle\text{Proj}_\Theta(w) - w, \text{Proj}_\Theta(w) - x\rangle \leq 0.
$$

Therefore,

$$
\langle z^i(t), x^i - \eta_t g^i(t) - x\rangle = \langle\text{Proj}_\Theta(x^i(t) - \eta_t g^i(t)) - (x^i(t) - \eta_t g^i(t)), x^i(t) - \eta_t g^i(t) - x\rangle \leq 0.
$$

Applying this inequality in (28) and using (19) we find

$$\frac{1}{\eta_t}\langle \bar{z}(t), \bar{x}(t) - x\rangle \leq \frac{1}{n\eta_t}\sum_{i=1}^{n}\langle z^i(t), (\bar{x}(t) - x^i(t)) + \eta_t g^i(t)\rangle$$

$$\leq \frac{1}{n\eta_t}\sum_{i=1}^{n}\left\|z^i(t)\right\|\left\|x^i(t) - \bar{x}(t)\right\| + \frac{1}{n}\sum_{i=1}^{n}\left\|z^i(t)\right\|\left\|g^i(t)\right\|$$

$$\leq \frac{1}{2n\eta_t}\sum_{i=1}^{n}\left[\frac{\eta_t^2\left\|g^i(t)\right\|^2}{1-\rho} + (1-\rho)\left\|x^i - \bar{x}(t)\right\|^2\right] + \frac{\eta_t}{n}\sum_{i=1}^{n}\left\|g^i(t)\right\|^2$$

$$\leq \frac{3\eta_t}{2(1-\rho)n}\sum_{i=1}^{n}\left\|g^i(t)\right\|^2 + \frac{1-\rho}{2n\eta_t}\sum_{i=1}^{n}\left\|x^i(t) - \bar{x}(t)\right\|^2. \tag{29}$$

Inserting (29) and (27) in (23) and using the fact that $\eta_t = \frac{2}{\alpha t}$ we get

$$\mathbb{E}[f(\bar{x}(t)) - f(x)|\mathcal{F}_t] \leq \frac{1}{2\eta_t}\mathbb{E}[a_t - a_{t+1}|\mathcal{F}_t] - \frac{\alpha}{4}a_t$$

$$+ \frac{(1+4\bar{L})t\alpha(1-\rho)}{4n}\sum_{i=1}^{n}\left\|x^i - \bar{x}(t)\right\|^2 +$$

$$+ \frac{7\eta_t}{2(1-\rho)n}\sum_{i=1}^{n}\mathbb{E}[\left\|g^i(t)\right\|^2|\mathcal{F}_t] + \frac{(\kappa_\beta L)^2}{\alpha}dh_t^{2(\beta-1)} + \frac{\bar{L}\mathcal{K}^2}{4t\alpha(1-\rho)}.$$

where the last inequality follows from. Taking the expectations, setting $r_t := \mathbb{E}[a_t]$ and applying Lemma 2 we get

$$\mathbb{E}\left[f(\bar{x}(t)) - f(x)\right] \leq \frac{r_t - r_{t+1}}{2\eta_t} - \frac{\alpha}{4}r_t + \frac{(1+4\bar{L})t\alpha(1-\rho)}{4}\Delta(t) + \tag{30}$$

$$+ \frac{7}{\alpha(1-\rho)t}\left(9\kappa G^2 + d\left(\frac{9h_t^2\kappa\bar{L}^2}{8} + \frac{3\kappa\sigma^2}{2h_t^2}\right)\right)$$

$$+ \frac{(\kappa_\beta L)^2}{\alpha}dh_t^{2(\beta-1)} + \frac{\bar{L}\mathcal{K}^2}{4t\alpha(1-\rho)}.$$

Notice that for our choice of $\eta_t = \frac{2}{\alpha t}$ we have

$$\sum_{t=1}^{T_0}\left(\frac{r_t - r_{t+1}}{2\eta_t} - \frac{\alpha}{4}r_t\right) \leq 0.$$

Recall that $h_t = t^{-\frac{1}{2\beta}}$. We can see now that this choice of $h_t$ is the minimizer of the main term depending on $h_t$ on the right hand side of (30), which is (up to multiplicative constant) of the order of $h_t^{2(\beta-1)} + \frac{1}{th_t^2}$. By substituting this $h_t$ in (30) and summing over $t$ we get

$$\sum_{t=1}^{T_0}\mathbb{E}\left[f(\bar{x}(t)) - f(x)\right] \leq \frac{(1+4\bar{L})\alpha(1-\rho)}{4}\sum_{t=1}^{T_0}t\Delta(t) + \mathcal{B}_1\frac{d}{\alpha(1-\rho)}T_0^{\frac{1}{\beta}} + \frac{\bar{L}\mathcal{K}^2}{4\alpha(1-\rho)}\left(\log(T_0) + 1\right),$$

where $\mathcal{B}_1 = 7\beta\left(\frac{9\kappa G^2}{d} + (\frac{9\kappa\bar{L}^2}{8} + \frac{3\kappa\sigma^2}{2})\right) + \beta(\kappa_\beta L)^2$. Finally, using Lemma 3 we obtain

$$\sum_{t=1}^{T_0}\mathbb{E}\left[f(\bar{x}(t)) - f(x)\right] \leq \mathcal{B}_1\frac{d}{\alpha(1-\rho)}T_0^{\frac{1}{\beta}} + \mathcal{B}_2\frac{\rho^2}{1-\rho}\frac{d}{\alpha}T_0^{\frac{1}{\beta}} + \frac{\mathcal{B}_3}{\alpha(1-\rho)}\left(\log(T_0) + 1\right),$$

where $\mathcal{B}_2 = \frac{\beta(1+4\bar{L})}{4}\mathcal{A}$, and $\mathcal{B}_3 = \bar{L}\mathcal{K}^2$. This proves the first bound of the theorem. The second bound (7) follows immediately by the convexity of $f$. $\qquad\square$

**Corollary 5.** *Let Assumptions A, B, and C hold. Let $\Theta$ be a convex compact subset of $\mathbb{R}^d$. Assume that $diam(\Theta) \leq \mathcal{K}$ and $\max_{x\in\Theta}\|\nabla f(x)\| \leq G$. If the updates $x^i(t)$ are defined by Algorithm*

*1, in which the gradient estimators for $i$-th agent are defined by Algorithm 2 with $F = f_i$, $i = 1, \ldots, n$, and parameters $\eta_t = \frac{4}{\alpha(t+1)}, h_t = t^{-\frac{1}{2\beta}}$ then the local average estimator $\hat{x}^i(T_0) = \frac{2}{T_0(T_0+1)} \sum_{t=1}^{T_0} t x^i(t)$ satisfies*

$$\mathbb{E}[\|\hat{x}^i(T_0) - x^*\|^2] \leq \mathcal{C} \min\left\{1, \frac{d}{\alpha^2(1-\rho)} T_0^{-\frac{\beta-1}{\beta}} \left(1 + \frac{n\rho^2}{(1-\rho)T_0}\right)\right\}, \quad i = 1, \ldots, n,$$

*where $\mathcal{C} > 0$ is a positive constant independent of $T_0, d, \alpha, n, \rho$.*

**Proof.** In contrast to the previous proofs, now we have $\eta_t = \frac{4}{\alpha(t+1)}$ rather than $\eta_t = \frac{2}{\alpha t}$.

1°. Inspection of the proof of Lemma 3 immediately yields that Lemma 3 remains valid with $\eta_t = \frac{4}{\alpha(t+1)}$ instead of $\eta_t = \frac{2}{\alpha t}$, up to a change in constant $\mathcal{A}$. Thus,

$$\Delta(t) \leq \bar{A} \left(\frac{\rho}{1-\rho}\right)^2 \frac{d}{\alpha^2} t^{-\frac{2\beta-1}{\beta}}, \tag{31}$$

$$\mathbb{E}\left[\|\hat{x}^i(t) - \bar{x}(t)\|^2\right] \leq \bar{A} n \left(\frac{\rho}{1-\rho}\right)^2 \frac{d}{\alpha^2} t^{-\frac{2\beta-1}{\beta}}, \quad i = 1, \ldots, n, \tag{32}$$

where $\bar{A} > 0$ is a constant independent of $t, d, \alpha, n, \rho$.

2°. Next, we show that, up to changes in constants $\mathcal{B}_i$, the bound (7) of Theorem 4 remains valid with $\eta_t = \frac{4}{\alpha(t+1)}$ instead of $\eta_t = \frac{2}{\alpha t}$ if we replace $\hat{x}(T_0)$ in (7) by the estimator

$$\hat{x}_\star(T_0) := \frac{2}{T_0(T_0+1)} \sum_{t=1}^{T_0} t\bar{x}(t).$$

Indeed, repeating the proof of Theorem 4 until (30), multiplying both sides of (30) by $t$, summing up from $t = 1$ to $T_0$ and using the fact that

$$\sum_{t=1}^{T_0} \left(\frac{t(r_t - r_{t+1})}{2\eta_t} - \frac{\alpha}{4} t r_t\right) \leq 0 \qquad \text{if } \eta_t = \frac{4}{\alpha(t+1)},$$

we find that, for all $x \in \Theta$,

$$\sum_{t=1}^{T_0} t\, \mathbb{E}\left[f(\bar{x}(t)) - f(x)\right] \leq \frac{(1+4\bar{L})\alpha(1-\rho)}{4} \sum_{t=1}^{T_0} t^2 \Delta(t) + \bar{\mathcal{B}}_1 \frac{d}{\alpha} T_0^{1+\frac{1}{\beta}} + \frac{\bar{L}\mathcal{K}^2}{4\alpha(1-\rho)} T_0,$$

where $\bar{\mathcal{B}}_1$ is a positive constant independent of $T_0, d, \alpha, n, \rho$. Using (31) we get, for all $x \in \Theta$,

$$\frac{2}{T_0(T_0+1)} \sum_{t=1}^{T_0} t\, \mathbb{E}\left[f(\bar{x}(t)) - f(x)\right] \leq \bar{\mathcal{B}}_2 \frac{d}{\alpha(1-\rho)} T_0^{-1+\frac{1}{\beta}},$$

where $\bar{\mathcal{B}}_2$ is a positive constant independent of $T_0, d, \alpha, n, \rho$. In view of the convexity of $f$, it follows that

$$\mathbb{E}\left[f(\hat{x}_\star(T_0)) - f(x^*)\right] \leq \bar{\mathcal{B}}_2 \frac{d}{\alpha(1-\rho)} T_0^{-1+\frac{1}{\beta}}.$$

As $f$ is strongly convex we also have

$$\mathbb{E}\left[\|\hat{x}_\star(T_0) - x^*\|^2\right] \leq 2\bar{\mathcal{B}}_2 \frac{d}{\alpha^2(1-\rho)} T_0^{-1+\frac{1}{\beta}}. \tag{33}$$

On the other hand, convexity of function $\|\cdot\|^2$ implies that

$$\|\hat{x}^i(T_0) - \hat{x}_\star(T_0)\|^2 = \left\|\frac{2}{T_0(T_0+1)} \sum_{t=1}^{T_0} t(x^i(t) - \bar{x}(t))\right\|^2$$

$$\leq \frac{2}{T_0(T_0+1)} \sum_{t=1}^{T_0} t\|x^i(t) - \bar{x}(t)\|^2. \tag{34}$$

Combining (32) and (34) we obtain

$$\mathbb{E}\big[\|\hat{x}^i(T_0) - \hat{x}_\star(T_0)\|^2\big] \leq \bar{\mathcal{C}}n\left(\frac{\rho}{1-\rho}\right)^2 \frac{d}{\alpha^2}T_0^{-\frac{2\beta-1}{\beta}}, \tag{35}$$

where $\bar{\mathcal{C}} > 0$ is a constant independent of $T_0, d, \alpha, n, \rho$. The desired result now follows from (33), (35) and the fact that $\|\hat{x}^i(T_0) - x^*\|$ is trivially bounded by the diameter of $\Theta$.

$\square$

## D  Proofs for Section 7

We first restate the following three lemmas from Akhavan et al. [2020].

**Lemma 9.** *Let for $\beta = 2$, Assumptions B and D hold. Let $\bar{g}(t)$ be the average of gradient estimators for $n$ agents defined each by (12), and $h = h_t$. If $\max_{x \in \Theta} \|\nabla f_i(x)\| \leq G$, for $1 \leq i \leq n$, then*

$$\mathbb{E}[\|\bar{g}(t)\|^2] \leq 9\kappa\left(G^2 d + \frac{L^2 d^2 h_t^2}{2}\right) + \frac{3\kappa d^2 \sigma^2}{2h_t^2}.$$

Introduce the notation

$$\hat{f}_t(x) = \mathbb{E}f(x + h_t \tilde{\zeta}), \qquad \forall x \in \mathbb{R}^d,$$

and

$$\hat{f}_t^i(x) = \mathbb{E}f_i(x + h_t \tilde{\zeta}), \qquad \forall x \in \mathbb{R}^d.$$

**Lemma 10.** *Suppose $f_i$ is differentiable. For the conditional expectation given $\mathcal{F}_t$, we have*

$$\mathbb{E}[g^i(t)|\mathcal{F}_t] = \nabla \hat{f}_t^i(x^i(t)).$$

**Lemma 11.** *If $f$ is $\alpha$-strongly convex then $\hat{f}_t$ is $\alpha$-strongly convex. If $f \in \mathcal{F}_2(L)$, for any $x \in \mathbb{R}^d$ and $h_t > 0$, we have*

$$|\hat{f}_t(x) - f(x)| \leq Lh_t^2,$$

*and*

$$|\mathbb{E}f(x \pm h_t \zeta_t) - f(x)| \leq Lh_t^2.$$

**Lemma 12.** *Let Assumptions A, B, and D hold with $\beta = 2$. Let $\Theta$ be a convex compact subset of $\mathbb{R}^d$, and assume that $diam(\Theta) \leq \mathcal{K}$. Assume that $\max_{x \in \Theta} \|\nabla f_i(x)\| \leq G$, for $1 \leq i \leq n$. Let the updates $x^i(t), \bar{x}(t)$ be defined by Algorithm 1, in which the gradient estimator for $i$-th agent is defined by (12), and $\eta_t = \frac{1}{\alpha t}$, $h_t = \left(\frac{3d^2\sigma^2}{2L\alpha t + 9L^2 d^2}\right)^{1/4}$. Then*

$$\Delta(t) \leq \left(\frac{\rho}{1-\rho}\right)^2\left(\mathcal{A}_1' \frac{d}{\alpha^{3/2}}t^{-\frac{3}{2}} + \mathcal{A}_2' \frac{d^2}{\alpha^2}t^{-2}\right),$$

*where $\mathcal{A}_1'$ and $\mathcal{A}_2'$ are positive constants independent of $T, d, \alpha, n, \rho$.*

*Proof.* Similarly to Lemma 3 we obtain

$$\mathbb{E}[V(t+1)|\mathcal{F}_t] \leq \rho^2(1+2\lambda)V(t) + \rho^2(4+\frac{2}{\lambda})\eta_t^2 \sum_{i=1}^n \mathbb{E}[\|g^i(t)\|^2 |\mathcal{F}_t].$$

Choosing $\lambda = \frac{1-\rho}{2\rho}$ and using Lemma 9 we get

$$\mathbb{E}[V(t+1)|\mathcal{F}_t] \leq \rho V(t) + \frac{4\rho^2}{1-\rho}\eta_t^2\left(9(G^2 d + \frac{L^2 d^2 h_t^2}{2}) + \frac{3d^2\sigma^2}{2h_t^2}\right).$$

Taking here the expectations and setting $\eta_t = \frac{1}{\alpha t}$ and $h_t = \left(\frac{3d^2\sigma^2}{2L\alpha t + 9L^2 d^2}\right)^{1/4}$ yields

$$\Delta(t+1) \leq \rho\Delta(t) + \frac{\rho^2}{1-\rho}\left(\mathcal{A}_3' \frac{d}{\alpha^{3/2}t^{3/2}} + \mathcal{A}_4' \frac{d^2}{\alpha^2 t^2}\right)$$

with $\mathcal{A}'_3 = 2\sqrt{6L}\sigma$, and $\mathcal{A}'_4 = 12\sqrt{3}L\sigma + \frac{36G^2}{d}$. On the other hand, by recursion we have

$$\Delta(t+1) \le \rho^t \Delta(1) + \frac{\rho^2}{1-\rho} \frac{d}{\alpha^{3/2}} \left( \mathcal{A}'_3 \sum_{s=1}^t s^{-\frac{3}{2}} \rho^{t-s} + \mathcal{A}'_4 \frac{d}{\alpha^{1/2}} + \sum_{s=1}^t s^{-2} \rho^{t-s} \right).$$

Here $\Delta(1) = 0$ due to the initialization. The sums on right hand side can be estimated by using an argument, which is quite analogous to what was done in the proof of Lemma 3, after equation (22), leading to the result of the lemma. $\qquad\square$

**Lemma 13.** *Let the assumptions of Lemma 12 hold and let $f$ be an $\alpha$-strongly convex function. Then*

$$\mathbb{E}[\|\bar{x}(t) - x^*\|^2] \le \frac{\mathcal{C}}{1-\rho} \left( \frac{d}{t^{1/2}\alpha^{3/2}} + \frac{d^2}{t\alpha^2} \right),$$

*where $\mathcal{C} > 0$ is a constant independent of $T, d, \alpha, n, \rho$.*

*Proof.* First note that due to the strong convexity assumption we have

$$\|\bar{x}(1) - x^*\|^2 \le \frac{G^2}{\alpha^2}.$$

Therefore, for $t = 1$ the result holds. For $t \ge 2$, by the definition of the algorithm we have

$$\|\bar{x}(t+1) - x^*\|^2 \le \|\bar{x}(t) - x^*\|^2 + \eta_t^2 \|\bar{g}(t)\|^2 + \|\bar{z}(t)\|^2 - 2\eta_t \langle \bar{g}(t), \bar{z}(t) \rangle - \\ - 2\eta_t \langle \bar{g}(t), \bar{x}(t) - x^* \rangle + 2\langle \bar{x}(t) - x^*, \bar{z}(t) \rangle.$$

Taking conditional expectations we get

$$\mathbb{E}[a_{t+1}|\mathcal{F}_t] \le a_t + \frac{2\eta_t^2}{n} \sum_{i=1}^n \mathbb{E}[\|g^i(t)\|^2 |\mathcal{F}_t] - 2\eta_t \mathbb{E}[\langle \bar{g}(t), \bar{z}(t) \rangle |\mathcal{F}_t] - \qquad (36)$$

$$- 2\eta_t \mathbb{E}[\langle \bar{g}(t), \bar{x}(t) - x^* \rangle |\mathcal{F}_t] + 2\mathbb{E}[\langle \bar{x}(t) - x^*, \bar{z}(t) \rangle |\mathcal{F}_t], \qquad (37)$$

where we used the fact that $\|z^i(t)\| \le \eta_t \|g^i(t)\|$ for $1 \le i \le n$.

For the term $-2\eta_t \mathbb{E}[\langle \bar{g}(t), \bar{x}(t) - x^* \rangle |\mathcal{F}_t]$ in (36), we have

$$-2\eta_t \mathbb{E}[\langle \bar{g}(t), \bar{x}(t) - x^* \rangle |\mathcal{F}_t] \le -\frac{2\eta_t}{n} \sum_{i=1}^n \Big( \mathbb{E}[\langle g^i(t) - \nabla \hat{f}_t^i(x^i(t)), \bar{x}(t) - x^* \rangle |\mathcal{F}_t] + \qquad (38)$$

$$+ \langle \nabla \hat{f}_t^i(x^i(t)) - \nabla \hat{f}_t^i(\bar{x}(t)), \bar{x}(t) - x^* \rangle + \qquad (39)$$

$$+ \langle \nabla \hat{f}_t(\bar{x}(t)), \bar{x}(t) - x^* \rangle \Big) \qquad (40)$$

For the term in (38), by Lemma 10 we have

$$-\frac{2\eta_t}{n} \sum_{i=1}^n \mathbb{E}[\langle g^i(t) - \nabla \hat{f}_t^i(x^i(t)), \bar{x}(t) - x^* \rangle |\mathcal{F}_t] = 0.$$

For the term in (39), decoupling yields

$$-\frac{2\eta_t}{n} \sum_{i=1}^n \langle \nabla \hat{f}_t^i(x^i(t)) - \nabla \hat{f}_t^i(\bar{x}(t)), \bar{x}(t) - x^* \rangle \le \frac{\eta_t t\alpha}{n}(1-\rho)V(t) + \frac{\bar{L}^2 \eta_t}{t\alpha} \frac{1}{1-\rho} a_t.$$

Next, we use the strong convexity (cf. Lemma 11) to handle (40):

$$-2\eta_t \langle \nabla \hat{f}_t(\bar{x}(t)), \bar{x}(t) - x^* \rangle \le -2\eta_t \alpha a_t.$$

Finally, for the term containing $2\langle \bar{x}(t) - x^*, \bar{z}(t) \rangle$ in (37) we obtain similarly to (29) that

$$2\mathbb{E}[\langle \bar{x}(t) - x^*, \bar{z}(t) \rangle |\mathcal{F}_t] \le \frac{3\eta_t^2}{(1-\rho)n} \sum_{i=1}^n \mathbb{E}[\|g^i(t)\|^2 |\mathcal{F}_t] + \frac{1-\rho}{n}V(t).$$

Combining the above inequalities yields

$$\mathbb{E}[a_{t+1}|\mathcal{F}_t] \leq (1 - 2\eta_t\alpha)a_t + \frac{2\eta_t^2}{n}\sum_{i=1}^n \mathbb{E}[\|\bar{g}(t)\|^2 |\mathcal{F}_t] - 2\eta_t\mathbb{E}[\langle \bar{g}(t), \bar{z}(t)\rangle|\mathcal{F}_t] + \frac{\eta_t\bar{L}^2\mathcal{K}^2}{t\alpha(1-\rho)} +$$

$$+ \frac{\eta_t t\alpha + 1}{n}(1 - \rho)V(t) + \frac{3\eta_t^2}{(1-\rho)n}\sum_{i=1}^n \mathbb{E}[\|g^i(t)\|^2 |\mathcal{F}_t].$$

Now, recalling that $\eta_t = \frac{1}{t\alpha}$, $h_t = \left(\frac{3d^2\sigma^2}{2L\alpha t + 9L^2 d^2}\right)^{1/4}$, taking the expectations and applying Lemma 9 we find

$$r_{t+1} \leq \left(1 - \frac{2}{t}\right)r_t + 2(1 - \rho)\Delta(t) + \frac{C}{(1-\rho)}\left(\frac{d}{t^{3/2}\alpha^{3/2}} + \frac{d^2}{t^2\alpha^2}\right), \tag{41}$$

where $r_t = \mathbb{E}[a_t]$, and $C > 0$ is a constant independent of $T, d, \alpha, n, \rho$. Using Lemma 12 to bound $\Delta(t)$ in (41) we get

$$r_{t+1} \leq \left(1 - \frac{2}{t}\right)r_t + \frac{C'}{(1-\rho)}\left(\frac{d}{t^{3/2}\alpha^{3/2}} + \frac{d^2}{t^2\alpha^2}\right),$$

where $C' > 0$ is a constant independent of $T, d, \alpha, n, \rho$. The desired result follows from this recursion by applying [Akhavan et al., 2020, Lemma D.1]. □

**Theorem 7.** *Let $f$ be an $\alpha$-strongly convex function. Let Assumptions A, B, and D hold with $\beta = 2$. Let $\Theta$ be a convex compact subset of $\mathbb{R}^d$, and assume that $diam(\Theta) \leq \mathcal{K}$. Assume that $\max_{x\in\Theta} \|\nabla f_i(x)\| \leq G$, for $1 \leq i \leq n$. Let the updates $x^i(t), \bar{x}(t)$ be defined by Algorithm 1, in which the gradient estimator for $i$-th agent is defined by (12), and $\eta_t = \frac{1}{\alpha t}$, $h_t = \left(\frac{3d^2\sigma^2}{2L\alpha t + 9L^2 d^2}\right)^{1/4}$. Then for the estimator $\tilde{x}(T) = \frac{1}{T - \lfloor T/2 \rfloor}\sum_{t=\lfloor T/2 \rfloor + 1}^T \bar{x}(t)$ we have*

$$\mathbb{E}[f(\tilde{x}(T)) - f(x^*)] \leq \frac{\mathcal{B}}{1 - \rho}\left(\frac{d}{\sqrt{\alpha T}} + \frac{d^2}{\alpha T}\right),$$

*where $\mathcal{B} > 0$ is a constant independent of $T, d, \alpha, n, \rho$.*

*Proof.* Fix $x \in \Theta$. Due to the $\alpha$-strong convexity of $\hat{f}_t$, we have

$$\hat{f}_t(\bar{x}(t)) - \hat{f}_t(x^*) \leq \langle \nabla\hat{f}_t(\bar{x}(t)), \bar{x}(t) - x^*\rangle - \frac{\alpha}{2}\|\bar{x}(t) - x^*\|^2.$$

Thus, by Lemma 11 we get

$$f(\bar{x}(t)) - f(x^*) \leq 2Lh_t^2 + \langle \nabla\hat{f}_t(\bar{x}(t)), \bar{x}(t) - x^*\rangle - \frac{\alpha}{2}\|\bar{x}(t) - x^*\|^2.$$

Let $a_t = \|\bar{x}(t) - x^*\|^2$. Taking conditional expectations and applying Lemma 10 we obtain

$$\mathbb{E}[f(\bar{x}(t)) - f(x^*)|\mathcal{F}_t] \leq 2Lh_t^2 + \frac{1}{n}\sum_{i=1}^n \langle\nabla\hat{f}_t^i(\bar{x}(t)) - \nabla\hat{f}_t^i(x^i(t)), \bar{x}(t) - x^*\rangle - \frac{\alpha}{2}a_t$$

$$+ \mathbb{E}[\langle\bar{g}(t), \bar{x}(t) - x^*\rangle|\mathcal{F}_t]$$

$$\leq 2Lh_t^2 + \frac{1}{n}\sum_{i=1}^n \mathbb{E}[\langle\nabla\hat{f}_t^i(\bar{x}(t)) - \nabla\hat{f}_t^i(x^i(t)), \bar{x}(t) - x^*\rangle|\mathcal{F}_t]$$

$$- \frac{\alpha}{2}a_t + \frac{a_t - \mathbb{E}[a_{t+1}|\mathcal{F}_t]}{2\eta_t}$$

$$+ \frac{1}{\eta_t}\mathbb{E}[\langle\bar{z}(t), \bar{x}(t) - x^*\rangle|\mathcal{F}_t] + \frac{2\eta_t}{n}\sum_{i=1}^n \mathbb{E}[\|g^i(t)\|^2 |\mathcal{F}_t], \tag{42}$$

where the last inequality uses the definition of the algorithm. Now, by decoupling we find

$$\frac{1}{n}\sum_{i=1}^n \langle\nabla\hat{f}_t^i(\bar{x}(t)) - \nabla\hat{f}_t^i(x^i(t)), \bar{x}(t) - x^*\rangle \leq \frac{t\alpha}{2n}(1 - \rho)V(t) + \frac{1}{2(1-\rho)}\frac{\bar{L}^2}{t\alpha}\mathcal{K}^2, \tag{43}$$

while similarly to (29) we also have

$$\frac{1}{\eta_t}\mathbb{E}[\langle \bar{z}(t), \bar{x}(t) - x^* \rangle | \mathcal{F}_t] \leq \frac{1}{1-\rho}\frac{3\eta_t}{2n}\sum_{i=1}^{n}\mathbb{E}[\|g^i(t)\|^2 | \mathcal{F}_t] + (1-\rho)\frac{1}{2n\eta_t}V(t). \qquad (44)$$

Combining the above inequalities and applying Lemma 9 yields

$$\mathbb{E}[f(\bar{x}(t)) - f(x^*)|\mathcal{F}_t] \leq \left(\frac{1}{\eta_t} + t\alpha\right)\frac{1-\rho}{2n}V(t) + \frac{1}{2(1-\rho)}\frac{\bar{L}^2}{t\alpha}\mathcal{K}^2 - \frac{\alpha}{2}a_t + \frac{a_t - \mathbb{E}[a_{t+1}|\mathcal{F}_t]}{2\eta_t} +$$

$$+ 2Lh_t^2 + \left(2 + \frac{3}{2(1-\rho)}\right)\frac{\eta_t}{n}\sum_{i=1}^{n}\mathbb{E}[\|g^i(t)\|^2 | \mathcal{F}_t]. \qquad (45)$$

Let $r_t = \mathbb{E}[a_t]$. Using the fact that $\eta_t = \frac{1}{\alpha t}$, $h_t = \left(\frac{3d^2\sigma^2}{2L\alpha t + 9L^2 d^2}\right)^{1/4}$, taking the expectations in (45) and applying Lemma 9 we find

$$\mathbb{E}[f(\bar{x}(t)) - f(x^*)] \leq t\alpha\left(\frac{r_t - r_{t+1}}{2}\right) - \frac{\alpha}{2}r_t + (1-\rho)\alpha t\Delta(t) + \frac{C_1}{1-\rho}\left(\frac{d}{\sqrt{\alpha t}} + \frac{d^2}{\alpha t}\right),$$

where $C_1 > 0$ is a constant independent of $T, d, \alpha, n, \rho$. Summing up both sides over $t$ gives

$$\sum_{t=\lfloor \frac{T}{2}\rfloor+1}^{T}\mathbb{E}[f(\bar{x}(t)) - f(x^*)] \leq r_{\lfloor \frac{T}{2}\rfloor+1}\frac{\lfloor \frac{T}{2}\rfloor\alpha}{2} + (1-\rho)\alpha\sum_{t=\lfloor \frac{T}{2}\rfloor+1}^{T}t\Delta(t) + \frac{C_2}{1-\rho}\left(\frac{d\sqrt{T}}{\sqrt{\alpha}} + \frac{d^2}{\alpha}\right)$$

where $C_2 > 0$ is a constant independent of $T, d, \alpha, n, \rho$. We now apply Lemma 12 to bound $\Delta(t)$ and Lemma 13 to bound $r_{\lfloor \frac{T}{2}\rfloor+1}$. It follows that

$$\sum_{t=\lfloor \frac{T}{2}\rfloor+1}^{T}\mathbb{E}[f(\bar{x}(t)) - f(x^*)] \leq \frac{C_3}{1-\rho}\left(\frac{d\sqrt{T}}{\sqrt{\alpha}} + \frac{d^2}{\alpha}\right),$$

where $C_3 > 0$ is a constant independent of $T, d, \alpha, n, \rho$. The desired bound for $\mathbb{E}[f(\tilde{x}(T)) - f(x^*)]$ follows from this inequality by the convexity of $f$.

$\square$

# E  Numerical Experiments

In this section we present a numerical comparison between the proposed method and the zero-order method in Akhavan et al. [2020] based on 2-point gradient estimator. Since the goal is to study the effect of the new gradient estimator, we consider the standard (undistributed) setting.

We wish to minimize the following function $f : \mathbb{R}^d \to \mathbb{R}$,

$$f(x) = \frac{\alpha}{2}x^\top Ax + Lh^3\sum_{i=1}^{d}\psi(h^{-1}x_i), \qquad (46)$$

where $\alpha, L, h$ are positive parameters, $A$ is a positive definite matrix in $\mathbb{R}^{d\times d}$ with smallest eigenvalue equal to 1, and $\psi(x) = \int_{-\infty}^{x}\int_{-\infty}^{z}\phi(t)dtdz$, with

$$\phi(x) = \begin{cases} 0 & \text{if } x < -a \\ \frac{2}{a}x + 2 & \text{if } -a \leq x < -\frac{a}{2} \\ -\frac{2}{a}x & \text{if } -\frac{a}{2} \leq x \leq \frac{a}{2} \\ \frac{2}{a}x - 2 & \text{if } \frac{a}{2} \leq x \leq a \\ 0 & \text{if } a < x, \end{cases}$$

where $a > 0$. A direct computation gives that

$$\psi(x) = \begin{cases} 0 & \text{if } x < -a \\ \frac{x^3}{3a} + ax^2 + ax + \frac{a^2}{3} & \text{if } -a \leq x < -\frac{a}{2} \\ -\frac{x^3}{3a} + \frac{a}{2}x + \frac{a^2}{4} & \text{if } -\frac{a}{2} \leq x \leq \frac{a}{2} \\ \frac{x^3}{3a} - ax^2 + ax + \frac{a^2}{6} & \text{if } \frac{a}{2} \leq x \leq a \\ \frac{a^2}{2} & \text{if } a < x. \end{cases}$$

Let $\Theta = \{x \in \mathbb{R}^d : \|x\| \le 1, \text{ and } x_i \le 0, \text{ for } 1 \le i \le d\}$. Since for any $x \in \Theta$, $\phi(x) \ge 0$, then $\psi$ is convex on $\Theta$, which implies $\alpha$-strong convexity of $f$ on $\Theta$. Also, the second derivative of $Lh^3\psi(h^{-1}x)$ is Lipschitz continuous with Lipschitz constant equal to $\frac{2L}{a}$. Therefore $f$ is $\beta$-Hölder with $\beta = 3$. We choose the kernel function, $K : [-1,1] \to \mathbb{R}$, such that $K(x) = \frac{15}{8}x(5 - 7x^3)$. For each iteration $t$, we fix $h_t = t^{-\frac{1}{6}}$, and $\eta_t = \frac{2}{\alpha t}$. Function evaluations at a fixed point $x \in \mathbb{R}^d$ are obtained in the form $f(x) + \zeta$ where $\zeta$ is a random variable uniformly distributed in $[-5, 5]$.

In this implementation we assign $\alpha = 2$, $h = 10^{-3}$, $L = 10^{7.5}$, $a = 10$. We also let $A = B + \mathbb{I}$, where $B$ is a randomly generated sparse positive definite matrix in $\mathbb{R}^{d \times d}$ and $\mathbb{I}$ is the $d$-dimensional identity matrix. For the initialization, we generate a $d$-dimensional Gaussian random variable and project it on $\Theta$.

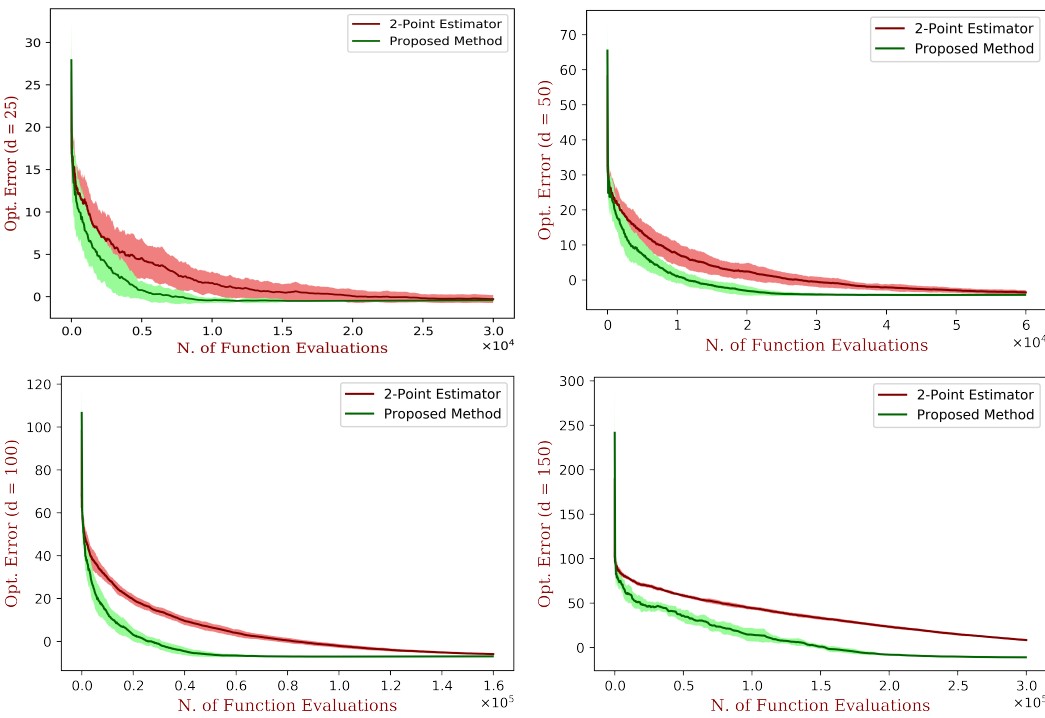

Figure 1: Optimization error vs. number of function evaluations for the 2-Point Estimator in Akhavan et al. [2020] and our method, run on function (46) for different number of variables ($d = 25, 50, 100, 150$ clockwise from top-left).

The design of $f$ in (46) is inspired by the function that has been used in the proof of the lower bound in Akhavan et al. [2020]. It is a quadratic function plus the perturbation $Lh^3 \sum_{i=1}^d \psi(h^{-1}x_i)$, which adds difficulty to estimation of the minimizer. We have chosen this worst case function to provide a comparison between two algorithms in a long run and growing dimension. In Figure 1 we display the average optimization error of the method proposed in this paper and that of the 2-Point estimator from Akhavan et al. [2020] versus the total number of function evaluations, for different dimensions $d$. This result is averaged over 40 trials, corresponding to different random initialization, noisy function evaluations and randomization in the optimization procedures. We would like to emphasize that both methods are considered with the same budget of function evaluations, which means that the number of iterations for the two algorithms differ. Thus, if $T$ is the total number of function evaluations, the 2-point estimator makes $T/2$ iterations, while the proposed method makes only $T/(2d)$ iterations.