# OpenReview forum: "Distributed Zero-Order Optimization under Adversarial Noise"
_NeurIPS.cc/2021/Conference — NeurIPS 2021 Poster_

### Official Review · Reviewer_pQgq · 2021-07-15

**Rating:** 7
**Confidence:** 4

**Summary:**

This paper studies zero-order optimization for strongly convex functions. The authors propose a novel 2d point zero-order gradient estimator, combing with distributed projected gradient descent, they provide upper bounds that are robust to adversary noises and have better dependence on (1 - \rho). In addition, they provide an improved analysis for 2-Holder functions using 2 point gradient estimator in that the dimension dependence is improved from d^1.5 to d.

**Limitations And Societal Impact:**

Yes.

**Main Review:**

1. The paper proposes a novel 2d point zero-order gradient estimator, which is computationally more efficient by a factor (1/d^2) in terms of total calls of random variable generators during algorithm execution,  especially beneficial in high-dimension regimes.

2. The paper shows optimization upper bound that has network dependence (1 - \rho), improving from the previous (1 - \rho)^2, where \rho is the spectral gap of the mixing matrix. Although I’d appreciate it if the authors could provide a brief analysis into this dependence matter, I only find explicit dependence of (1 - \rho)^2 in reference [27], while the authors also mention this dependence in references [35][37]. Also, I notice that those mentioned references are considering unconstrained optimization while this paper uses convex compact constraint, could the authors discuss a little bit on this when comparing the network dependence?

3. The numerical experiment is pretty artificial, can we find some real but may simple machine learning tasks to demonstrate the effectiveness of the proposed algorithms?

4. I skipped the proof for section 7, for the parts I have read,
    4.1. In line 181, I am trying to understand why these values of \eta_t, and h_t lead to the best rates.
    4.2. In the proof of Lemma 2, we may half the constants since r has density 1/2.
    4.3. In appendix B, the statements for Lemma 1 and Lemma 2 seems to be reversed.
    4.4. In line 521, there are some obvious typos.
    4.5. In line 524, I am trying to understand when deriving B_1 why the dependence on 1/(1 - \rho) is dropped, the overall dependence on (1-\rho) in the lemma is not affected though.
    4.6. In equations below line 537, the last term misses T_0.

5. The paper title contains “adversarial noise”, however, it seems to me the noise model is a general unbiased one with bounded second moment, there is no adversary such as designed attack or arbitrary value manipulations, so I feel this adjective in title may not be that relevant.

6. The paper is in general well written and of high quality, the contribution is very solid.

Comments after rebuttal:
I thank all the replies from authors and other reviewers, and I keep my current assessment.


**Time Spent Reviewing:**

15

---

> ### Author Response · Authors · 2021-08-10
> **Response to Reviewer pQgq**
>
> We thank all reviewers for their valuable comments, which will be taken into account in the revision of the paper.
>
> \emph{Dependency on $\rho$ in references [35], and [37].} In [35] (Eq (4), page 6) you can find the definition of $\rho$, which is the same as ours. The dependency on $\rho$ is outlined in Theorems 1,2,3, and 4.
>
> Reference [37] studied noisy-free zeroth order distributed optimization. We did not discuss the dependency on $\rho$ in this paper.
>
> \emph{Comparison between unconstrained and constrained optimization.} We provide this comparison, since we believe that the translation of the methods from unconstrained
> to constrained optimization, does not affect the dependency on $\rho$. We will discuss and outline this fact in the revision.
>
> \emph{Providing experiments on real machine learning tasks.} Our numerical experiment in Appendix E is designed to test Algorithm 2 on "least favorable" functions arising in the lower bound. We agree that an experiment on a real-world application would be of interest. We will consider adding real data experiment to the revised version, but we feel this should go in the supplementary material as the main focus of the paper is theoretical.
>
> \emph {The assigned $\eta_t$ and $h_t$ lead to the best rate.} Considering line 552, eq 30, the assigned $h_t$ and $\eta_t$ are the minimizers of the rhs with respect to $d, t, \alpha$. We will emphasize this in the main body, for the revision.
>
> \emph{The typos and the mistake with the constant $\mathcal{B}_1$.} Thank you for noting and mentioning these. We agree and we will incorporate these in the revision.
>
> \emph{"Adversarial noise" and the title.} There is not a standard definition of an adversarial setting. In this paper's noise model, there is no i.i.d. assumption and the noise also can be non-stochastic. Therefore, at each step the imposed noise can be conditioned on the previous outputs of the algorithm and acts as an adversarial setting.

---

### Official Review · Reviewer_RUCN · 2021-07-15

**Rating:** 7
**Confidence:** 3

**Summary:**

This paper studies a distributed optimization problem where the nodes conform to a given adjacency graph. The prevailing formulation of this problem in the literature is based on first order (gradient based) algorithms. This paper however, following [11, 27, 28, 29, 35, 37] considers a zeroth order optimization problem where only direct observations from the function value are accessible (instead of gradient). The average objective function is assumed to be strongly convex and the local objectives are assumed to be Hölder continuous. A distributed optimization algorithm based on a novel gradient estimator is presented. It is shown that the proposed algorithm is favorable over the existing ones in both convergence rate and computational complexity.

**Limitations And Societal Impact:**

The limitations of the work are discussed in Section 8.

**Main Review:**

Originality: The work is a combination of well-known techniques. Graph structured distributed optimization is a well-studied problem under the first order setting. This paper uses a kernel based gradient estimation for smooth functions to translate the zeroth order information to the first order ones (Algorithm 2). This gradient estimation step which seems the main contribution is also similar to the existing methods.

Quality: The results are technically sound.

Clarity: The paper is very well written and well organized.

Significance: The results seem to improve over the existing ones both in computational complexity and convergence rate. Although, to better understand the scope and novelty of the results, I have a couple of questions detailed below.

1. The gradient estimation method seems completely independent of the particular "graph structured distributed" optimization problem. Is it true that if the same gradient estimation is used for a standard centralized problem that also results in an improvement? If not, could you please specify why it results in an improvement in the "graph structured distributed" optimization?

In particular, it seems that the novelty is in the gradient estimation. Otherwise, when the zeroth order information is translated to the first order information, the rest follows the standard optimization techniques. I would like to better understand the scope of this better estimation of the gradient. Are algorithm 2 and Lemmas 1,2 novel fundamental results on the way to estimate gradient of a smooth function? In that case, I think this should be clarified in the paper. The improvement in the gradient estimation should also be quantified and reported in the paper.


2. The difference between Algorithm 1 in [2] and Algorithm 2 in this paper seems very subtle. It seems that the only difference is that while, in this paper, the canonical basis vectors are used for the change in the input of the function, in [2] an arbitrary vector on the sphere is used. This subtle difference appears to result in a considerable gain in the gradient estimation:

i) an $O(\sqrt{d})$ improvement in the bound on bias; Lemma 1 vs Lemma 2.3 of [2]

ii) an $O(d)$  improvement in the bound on variance; Lemma 2 vs Lemma 2.4 of [2]

- Could authors provide some intuition on the reason for these improvements given the algorithms are essentially the same?

- These improvements are reflected in the convergence rate. Is it fair to say these improvements are the only reason for the improved convergence rates?



A minor comment: there seems to be an error in lines 458, 465 where Lemma 2 is placed before the proof of Lemma 1.



**Time Spent Reviewing:**

6

---

> ### Author Response · Authors · 2021-08-10
> **Response to Reviewer RUCN**
>
> We thank all reviewers for their valuable comments, which will be taken into account in the revision of the paper.
>
> \emph{Is it true that if the same gradient estimation is used for a standard centralized problem that also results in an improvement?} Yes, we discussed this in Remark 2 at page 7.
>
> \emph{Are algorithm 2 and Lemmas 1,2 novel fundamental results on the way to estimate gradient of a smooth function?}
> Yes, these results are novel. The key difference from other related techniques is in introducing the kernel $K$ coordinate-wise. It leads to faster rate and to computational gain.
>
> \emph{Could authors provide some intuition on the reason for these improvements given the algorithms are essentially the same?} The main difference between the algorithms is that for Alg. 1 one needs to generate random points on the sphere while for Alg. 2 randomization is
>  coordinate-wise. As we show in the paper, this results in an improvement in the bias term in {Lemma 1}. Also, we have a tighter bound for the variance term in {Lemma 2}.
>
> \emph{These improvements are reflected in the convergence rate. Is it fair to say these improvements are the only reason for the improved convergence rates?} Yes, this improvement is the only reason for getting the better dependency of the convergence rate on the dimension that we obtain.

---

> > ### Comment · Reviewer_RUCN · 2021-09-06
> > **Update**
> >
> > Thanks for the response. I maintain my rating of the paper.

---

### Official Review · Reviewer_LtqT · 2021-07-16

**Rating:** 7
**Confidence:** 4

**Summary:**

This paper considers the problem of distributed zeroth-order optimization for a class of strongly convex functions. The authors propose a distributed zeroth-order optimization algorithm that only requires only functional evaluations of the objective functions, subject to a general noise model. The error bounds of the algorithm have also been established and improve upon the previous works. Lower bounds have also been discussed and they are nearly optimal.

**Main Review:**

The paper is very well written and easily readable. But the presentation of the paper could be improved:

1. As is well known that zeroth-order optimization is closely related to bandit optimization in online learning, so it would be better if the authors could provide some discussions about the connections between these two directions;

2. The simulation results are lacking. It is preferred to present some experiments on real-world applications, and comparisons with more related algorithms are welcome;

3. Is the dependence of the bounds on the dimension optimal?

4. More discussions about Definition 1 should be given. Why consider this type of function?



**Time Spent Reviewing:**

4

---

> ### Author Response · Authors · 2021-08-10
> **Response to Reviewer LtqT**
>
> We thank all reviewers for their valuable comments, which will be taken into account in the revision of the paper.
>
> \emph{Discussing the relation between zeroth-order optimization and bandit optimization in online learning.}
> Thanks for suggesting this, we will include a discussion of this point in the revision.
>
> \emph{The simulation results are lacking.} Our numerical experiment in Appendix E is designed to test Algorithm 2 on "least favorable" functions arising in the lower bound. We agree that an experiment on a real-world application would be of interest. We will consider adding real data experiment to the revised version, but we feel this should go in the supplementary material as the main focus of the paper is theoretical.
>
> \emph{Is the dependence of the bounds on the dimension optimal?} There is no evidence that our bound is optimal with respect to $d$. We discuss this in Remark 2 at page 7, by comparing our bound with the minimax lower bound established in [2].
>
>
> \emph{More discussions about Definition 1 should be given. Why consider this type of function?} Definition 1 is one of the common ways to describe the  higher order smoothness of a function. It was used previously in papers on zero-order optimization (see e.g. references [2] and [3]) where one can find discussion about its relevance.

---

> > ### Comment · Reviewer_LtqT · 2021-08-26
> > **Thank you for your response**
> >
> > Thank you for your response. I've read the rebuttal and the authors have addressed all of my concerns.

---

### Official Review · Reviewer_N8uL · 2021-07-16

**Rating:** 7
**Confidence:** 4

**Summary:**

This paper considers distributed optimization with zero-order information and under rather general setting of noise in the evaluation of function values. The authors propose a zero-order projected gradient descent algorithm, in which a novel zero-order gradient estimation scheme is used. An important contribution of this paper is to provide explicit upper bounds for the average cumulative regret and optimization error of the proposed algorithm. The obtained bound is clearly presented and show the advantage of the proposed algorithm over existing ones.

**Limitations And Societal Impact:**

There is no foreseeable negative societal impact of this work.

**Main Review:**

Originality: the proposed kernel-based $2d$-point gradient estimator seems new. It is key to achieving improved evaluation upper bound and handling non-zero-mean noise.

Quality: this paper is of high quality. All the technical results are supported with proof and comments. Insights and limitations of the obtained results are also discussed. It would be better if the authors could provide some insight on why the proposed algorithm attains an upper bound that is proportional to $(1-\rho)$ rather than $(1-\rho)^2$.

Clarity: the paper is written with high quality. The motivation, technical results, and limitations are clearly presented. It is a pleasure to read this paper. It would be nice if the authors could include some numerical results in the main body of the paper.

Significance: the obtained bound is new and significantly improves existing bounds even for standard (centralized) zero-order optimization.





**Time Spent Reviewing:**

3

---

> ### Author Response · Authors · 2021-08-10
> **Response to Reviewer N8uL**
>
> We thank all reviewers for their valuable comments, which will be taken into account in the revision of the paper.
>
>
> \emph{It would be better if the authors could provide some insight on why the proposed algorithm attains an upper bound that is proportional to $(1-\rho)$ rather than $(1-\rho)^2$.} This improvement is based on deriving a more accurate bound for the discrepancy term that arises in distributed setting. This is due to improved technique of the proof. In the revision, we will give a precise pointer to the place in the proof of Theorems 4 and 7 that lead to this improvement.
>
> \emph{It would be nice if the authors could include some numerical results in the main body of the paper.} We did not include the numerical study in the main body as the paper is already dense in theoretical results
> and there is unfortunately no space left. We moved the simulations in Appendix E in order to have enough space to discuss the details and the motivation of the numerical experiment.

---

### Decision · Program_Chairs · 2021-09-27

**Decision:**

Accept (Poster)

**Comment:**

The reviewers are all in agreement that this work ought to be accepted, and the largely-minor concerns were all resolved following the author feedback.  There were no major points of discussion that need highlighting in this meta-review, but I encourage the authors to carefully incorporate the reviewer feedback in the final version.  As one specific point, I do agree that the word "adversarial" could potentially cause confusion, especially in the title.  Although the word has many meanings, it could serve as a disadvantage if the title gives potential readers the wrong impression.  I leave it up to the authors whether to change the title, but ask that they consider the possibility carefully.